J Physiol 604.7 (2026) pp 2845–2866

2845

# Low-dose lithium supplementation promotes musculoskeletal and metabolic health in ovariectomized female mice

Bianca M. Marcella[1,2], Jessica L. Braun[1,2,3] (iD), Amélie A.T. Marais[1,2], Briana L. Hockey[1,2] (iD), Mia S. Geromella[1,2], Ryan W. Baranowski[1,2], Rene Vandenboom[1,2], Rebecca E.K. MacPherson[3,4] (iD) and Val A. Fajardo[1,2,3] (iD)

[1] *Department of Kinesiology, Brock University, St. Catharines, Ontario, Canada*
[2] *Centre for Bone and Muscle Health, Brock University, St. Catharines, Ontario, Canada*
[3] *Centre for Neurosciences, Brock University, St. Catharines, Ontario, Canada*
[4] *Department of Health Sciences, Brock University, St. Catharines, Ontario, Canada*

Handling Editors: Paul Greenhaff & Robert Musci

The peer review history is available in the Supporting Information section of this article (https://doi.org/10.1113/JP289990#support-information-section).

**Bianca** earned her bachelor of science in biomedical sciences from Brock University in 2022. During her undergraduate degree, she was awarded an NSERC Undergraduate Student Research Award to study the effects of low-dose lithium supplementation and exercise on skeletal muscle strength in a mouse model of muscular dystrophy. She went on to complete a master of science in applied health sciences at Brock University, supported by a CIHR Canada Graduate Scholarship (CGS-M). Her graduate research investigated the effects of low-dose lithium supplementation on skeletal muscle strength and whole-body metabolism in ovariectomized mice, advancing the understanding of musculoskeletal and metabolic health in postmenopausal animal models. Bianca is currently pursuing her Doctor of Medicine degree at Queen's University, with an expected graduation in 2028. Bianca has continued interests in exercise physiology, women's health and metabolic regulation.

*The Journal of Physiology*

**Abstract figure legend** Low-dose lithium supplementation in ovariectomized mice enhances skeletal muscle contractility (isometric force and fatigue resistance), SERCA function and promotes favourable transcriptional reprogramming, while increasing bone density and modestly improving insulin sensitivity.

**Abstract** Postmenopausal women have an increased risk for age-related conditions like sarcopenia, osteoporosis and type 2 diabetes. Here we examined the effects of low-dose lithium (Li) supplementation, a well-known glycogen synthase kinase 3 (GSK3) inhibitor, on ovariectomized (OVX) mice, focusing on muscle strength and endurance, bone mineral density (BMD) and insulin sensitivity. 28-week-old female sham and OVX C57BL/6J mice were divided into three groups: sham, OVX and OVX-Li, with the latter receiving 50 mg/kg/day of Li in their drinking water for 8 weeks. Li supplementation enhanced isometric specific force and fatigue resistance of the soleus and extensor digitorum longus (EDL) muscles. Potential cellular mechanisms underlying these benefits include enhanced $Ca^{2+}$ uptake, lowered oxidative stress, increased expression of select mitochondrial markers and a blunted muscle transcriptomic response to OVX surgery with Li supplementation. OVX mice exhibited lower BMD than sham controls; however Li supplementation restored BMD levels. Finally Li supplementation yielded modest improvements in insulin tolerance. In conclusion our findings highlight the advantages of low-dose Li for musculoskeletal and metabolic health in OVX female mice.

(Received 22 August 2025; accepted after revision 16 December 2025; first published online 8 January 2026)
**Corresponding author** V. A. Fajardo: Canada Research Chair (Tier II), Department of Kinesiology, Brock University, St. Catharines, ON, Canada. Email: vfajardo@brocku.ca

**Key points**

- Ovariectomy surgery compromises musculoskeletal and metabolic health in female mice.
- Here, we explored the potential benefits of low-dose lithium (Li) treatment.
- Li improved muscle isometric force production and fatigue resistance.
- RNA-Seq demonstrated that Li blunted the effects of OVX surgery.
- Li supplementation increased bone mineral density and insulin tolerance.

## Introduction

Postmenopausal women experience alterations in body composition, skeletal muscle function and metabolism due to the loss of the protective effects of oestrogen. These changes include reduced skeletal muscle mass, strength and endurance, increased body weight and adiposity and decreased bone mineral density (BMD). Altogether, these changes increase the risk of adverse health outcomes such as sarcopenia, obesity, osteoporosis and type 2 diabetes (Ahanchi et al., 2024; Finkelstein et al., 2008; Geraci et al., 2021; Opoku et al., 2023; Sipilä et al., 2020). Although hormone replacement therapy can alleviate many negative effects of low oestrogen levels in postmenopausal females, it might not be a suitable option for everyone due to factors such as age, time since menopause and pre-existing health conditions. Thus, other viable treatments must be explored. To this end, the ovariectomized (OVX) rodent model, specifically rat and mouse, has been widely used to study novel interventions.

In alignment with the effects of oestrogen deficiency observed in human females, the reduction in oestrogen levels in OVX rodents following ovariectomy surgery results in increased adiposity, decreased lean muscle mass and strength, reduced daily energy expenditure and glucose and insulin tolerance (Hayward et al., 2021; Kurachi et al., 1993; Marcella, Geromella et al., 2022; Moran et al., 2006; Nishio et al., 2019). This makes the OVX rodent model valuable for exploring therapeutic strategies aimed at alleviating the many adverse effects associated with oestrogen deficiency.

Glycogen synthase kinase 3 (GSK3) is a serine/threonine kinase first identified for its role in regulating glycogen synthase in muscle (Beurel et al., 2015). Since then, GSK3 has been implicated in the pathologies of metabolic disorders, neurological disorders, cancer, ageing, inflammatory and immune diseases and myopathies (Beurel et al., 2015; Frame & Cohen, 2001; Marcella, Copeland et al., 2022; Marcella et al., 2024; McCubrey et al., 2014). GSK3 exists in two

isoforms, GSK3$\alpha$ and GSK3$\beta$, with the latter being the most dominant across several tissues (Beurel et al., 2015; Markussen et al., 2018). It is a constitutively active enzyme, and its activity is regulated by phosphorylation, where Ser phosphorylation (Ser 21, GSK3$\alpha$; and Ser9, GSK3$\beta$) inhibits GSK3 activity (Gillespie et al., 2013). Lithium (Li) is a well-known inhibitor of GSK3 that can indirectly inhibit GSK3 by activating pathways that promote Ser phosphorylation and can also directly inhibit GSK3 via $Mg^{2+}$ competition, which is a cofactor of GSK3 (Jope, 2003). Li is often prescribed for bipolar disorder, but it must be carefully monitored to keep serum levels within the narrow therapeutic range of 0.5–1.0 mM (Hamstra et al., 2023). This is essential to avoid potential adverse effects on tissues, especially the kidneys, as chronic high-dose Li therapy has frequently been linked to renal toxicity (Davis et al., 2018).

In mice, however, Li treatment within the therapeutic range of 0.4–0.8 mM has also been shown to cause signs of nephrotoxicity (Nespital et al., 2021). Nonetheless, lower doses below the bipolar therapeutic range are effective in improving various health outcomes in mice (Choi et al., 2010; Geromella et al., 2023b; Kurgan, Bott et al., 2019; Nespital et al., 2021; Whitley et al., 2020). For example, others have found that low-dose Li supplementation (10 mg/kg/day; serum Li concentration of 0.02 mM) in male mice attenuated high-fat diet-induced obesity (Choi et al., 2010). Using the same dose, we have shown that low-dose Li supplementation in male wild-type mice increased specific force production of the soleus and extensor digitorum longus (EDL) muscles, fatigue resistance of the soleus and whole-body daily energy expenditure (Geromella et al., 2023b; Whitley et al., 2020). With respect to bone, Li is known to have osteoprotective effects (Bai et al., 2019; Kurgan, Bott et al., 2019; Vachhani et al., 2018), and we have shown that low-dose Li supplementation in male mice activates the Wnt/$\beta$-catenin signalling pathway, which promotes osteogenesis (Kurgan et al., 2019b). Additionally, in male *mdx* mice (the preclinical model for Duchenne muscular dystrophy), we have shown that a slightly higher yet still considered low, dose of Li given that it is still below the clinical therapeutic range (50 mg/kg/day; serum Li concentration of 0.15 mM), increased soleus and EDL isometric force production, soleus fatigue resistance, BMD and insulin sensitivity (Marcella et al., 2024). This relatively higher dose of Li was chosen to ensure GSK3 inhibition in the diseased state, while remaining under the toxicity threshold in mice.

However, it is important to note that most previous studies demonstrating the benefits of Li-mediated GSK3 inhibition have been conducted in male mice. The effects in female mice – particularly in ovariectomized females, which exhibit declines in muscle function, BMD, energy expenditure and glucose regulation – remain unclear. Therefore, in this study we investigated whether Li supplementation at 50 mg/kg/day, a dose we have previously used in male *mdx* mice, can mitigate the musculoskeletal and metabolic impairments observed in female OVX mice. To this end, we assessed skeletal muscle function, body composition (lean mass, fat mass and BMD), energy expenditure and glucose and insulin tolerance.

## Methods

### Ethical approval

All experimental procedures on sham and OVX female mice were approved in advance by the Brock University Animal Care Committee (AUP no.: 23-03-01) and carried out in accordance with the Canadian Council on Animal Care guidelines. OVX surgeries were performed at the Jackson Laboratory, an Association for Assessment and Accreditation of Laboratory Animal Care (AAALAC)-accredited facility, and all OVX procedures received approval from the Jackson Laboratory Institutional Animal Care and Use Committee.

### Animals and study design

C57BL/6J sham ($n = 10$) and ovariectomized (OVX; $n = 19$) female mice were purchased from the Jackson Laboratory (strain #000664, Bar Harbor, ME, USA). Surgeries were performed at 26 weeks of age at their facility, and animals arrived at Brock University at 27 weeks of age. Ovariectomies were conducted under aseptic and atraumatic conditions within a HEPA-filtered laminar flow environment. Mice were anaesthetized with inhaled isoflurane, and carprofen was administered as an analgesic. Ophthalmic ointment was applied to prevent corneal drying, and the absence of pedal and palpebral reflexes before incision confirmed anaesthetic depth. Bilateral ovariectomy was performed through small dorsal incisions; the ovaries, oviducts and associated fat pads were removed, and incisions were closed using absorbable sutures and wound clips. A topical local anaesthetic was applied to the incision site. Postoperative care included the subcutaneous administration of warm, sterile saline, continued thermal support and frequent monitoring until the animals were ambulatory. Mice were examined daily until the incision sites were fully healed (7 days), after which wound clips were removed before shipment.

Upon arriving at Brock University the mice acclimated to the animal facility for 1 week, allowing the OVX and sham mice to age to 28 weeks. After acclimation the OVX mice were randomly divided into a control or Li-treated group, giving rise to three groups in total: sham, OVX and OVX-Li ($n = 9$–10 per group). OVX-Li mice were provided a low dose of lithium chloride (LiCl)

(50 mg/kg/day) via their drinking water for 8 weeks, and the other groups were given the same type of water bottles without LiCl supplementation. For the duration of the study, mice were dual-housed, had *ad libitum* access to food and water and followed a standard 12 h light and dark schedule.

### Dual-energy X-ray absorptiometry scanning

The iNSiGHT Small Animal dual-energy X-ray absorptiometry (DXA) Scanner (Scintica Instrumentation, ON, Canada) was used to measure total body mass, fat mass, lean mass and BMD at the end of the 8 week treatment schedule. Mice were anaesthetized under vaporized isoflurane (induction: 5% in oxygen at a flow rate of 1–2 L/min) and then positioned in the DXA scanner, which is fitted with tubing to deliver vaporized isoflurane at a maintenance concentration of 0.25% in oxygen (flow rate: 1–2 L/min). After the scan ($\sim$35 s), mice were allowed to recover fully before being returned to their home cages.

### Promethion metabolic cages

Energy expenditure, respiratory exchange ratio (RER) and cage ambulation were measured using the Promethion Metabolic Screening System (Sable Systems Inc., Las Vegas, NV, USA) at week 4. The mice were housed with their cagemates in metabolic cages for 36 h. Consequently the values for energy expenditure and cage ambulation were divided by 2, whereas respiratory exchange ratio remained unaltered.

### Glucose and insulin tolerance test

During the 8th week of the study, a glucose tolerance test (GTT) and insulin tolerance test (ITT) were performed. The GTT was performed 48 h prior to the ITT. For the GTT, the mice underwent an overnight fast, and during the test the mice did not have access to food. Blood samples from the tail were collected at timepoints 0, 15, 30, 45, 60, 90 and 120 min after an intraperitoneal glucose injection of 2 g/kg body weight. Blood glucose levels were measured from the tail vein using a handheld glucometer (Freestyle Lite, Abbott). For the ITT, blood glucose levels were measured at 0, 15, 30, 45, 60, 90 and 120 min after an intraperitoneal injection of 0.75 U/kg of insulin. Mice were allowed access to their food and were monitored for signs of hypoglycaemic shock. Both GTT and ITT were carried out starting at 10:00 h.

### Tissue collection

After the 8 week intervention, mice were killed by exsanguination under vaporized isoflurane anaesthesia (5% in oxygen at a flow rate of 1–2 L/min). Adequate anaesthetic depth was confirmed by the absence of a pedal withdrawal (toe pinch) reflex prior to proceeding. One EDL and soleus muscle were excised for *in vitro* contractile analysis, and the other EDL and soleus muscle were collected and frozen in liquid nitrogen to be homogenized in 20:1 (w/v) homogenizing buffer (250 mM sucrose, 5 mM HEPES, 0.2 mM phenylmethylsulfonyl fluoride, 0.2% [w/v] $NaN_3$ in distilled $H_2O$, pH 7.5). Samples were stored at $-80°C$. All tissue collection occurred between 10:00 h and 14:00 h, with subjects sampled one at a time in alternating order (sham, OVX, OVX-Li).

### Skeletal muscle contractility

Soleus and EDL muscles were carefully dissected and mounted onto an Aurora Scientific contractile apparatus (model 305B&701B) to assess muscle force production and fatigue, as previously described (Whitley et al., 2020). Muscle length was first set to optimal length, determined as the length at which the maximal isometric twitch force was observed. Subsequently muscles were subjected to a force–frequency protocol consisting of submaximal and maximal isometric contractions (1, 20, 40 80 and 160 Hz stimulation for 350 ms). Peak isometric force (mN) was determined across the range of stimulation frequencies and subsequently normalized to muscle cross-sectional area (CSA), which was calculated using the following formula: $CSA = m/l^*d^*(L_f/L_o)$, where m is muscle mass (mg); l is muscle length (mm); d is mammalian skeletal muscle density (1.06 mg/mm$^3$) (Mendez J, 1960). Lf/Lo is the fibre length-to-muscle length ratio (0.44 for the EDL and 0.71 for the soleus muscle) (Brooks & Faulkner, 1988). The fatigue protocol consisted of a 70 Hz (450 ms) stimulation every 2 s for 160 s. Data were presented as a percentage of initial force throughout the fatigue protocol. After the fatigue protocol soleus and EDL muscles were weighed and then mounted in optimal cutting temperature (OCT) compound in liquid nitrogen-cooled isopentane for immunofluorescent fibre-type analysis.

### Western blotting

Western blotting was done to assess protein levels of peroxisome proliferator-activated receptor gamma coactivator 1-alpha (PGC1$\alpha$), cytochrome c oxidase subunit 4 (COXIV), citrate synthase (CS), 4-hydroxynonenal (4-HNE) and $\beta$-catenin, as previously described (Baranowski et al., 2023). Standard SDS-PAGE was performed using BioRad 4–15% TGX gels (4561086; BioRad, Hercules, CA, USA), and polyvinylidene difluoride (PVDF) membranes were used for all proteins (BioRad). All membranes were blocked with

Everyblot buffer (#12010020, BioRad) for 10 min at room temperature and incubated with their respective primary antibodies diluted in 5% milk or 3% BSA in TBST. The primary antibody for PGC1$\alpha$ (ST1204) was obtained from Millipore (Burlington, MA, USA). Antibodies for COXIV (4850) and $\beta$-catenin (8480) were obtained from Cell Signalling Technology (Danvers). The antibodies for CS (ab96600) and 4-HNE (ab46545) were obtained from Abcam (Cambridge, UK). After primary incubation membranes were washed thrice for 5 min with TBST and then incubated for 60 min with the corresponding anti-mouse horseradish peroxidase secondary antibody (7076, Cell Signalling). Membranes were then washed another three times with TBTS for 5 min each prior to imaging using a BioRad ChemiDoc Imager with Immobilon ECL Ultra Western HRP Substrate (WBKLS0500, Sigma-Aldrich). Images were analysed using BioRad ImageLab software (BioRad) and were normalized to total protein analysed on a Ponceau stain.

### Immunoprecipitation

Immunoprecipitation was performed to detect total and phosphorylated GSK3$\beta$ content in the soleus and the EDL. The primary antibodies for total GSK3$\beta$ (9315) and phosphorylated GSK3$\beta$ (5558) were obtained from Cell Signalling Technology (Danvers). To prepare the eluent 50 µL of SureBeads (1614023, BioRad) was pipetted into Eppendorf tubes, and then 50 µL of PBST was added to each tube. The tubes were then vortexed, placed on a magnetic bead rack and the liquid was discarded. This was performed thrice; 3 µL of total GSK3$\beta$ primary antibody was then added with PBST and was incubated for 10 min. The beads were magnetized again, and the supernatant was removed. The antibody-bead complex was washed thrice with PBST; 75 µg of muscle homogenate was then added to the antibody-bead complex and incubated for 1 h. The beads were magnetized again, and the supernatant was removed. The antibody-bead complex was washed thrice with PBST; 1× Laemeli buffer was added, and the samples were heated to 70°C for 10 min. The beads were magnetized, and the supernatant was removed and stored for future use. The western blotting protocol was performed as described above; however for this protocol TidyBlot was used as the secondary antibody (STAR209PT, BioRad) instead of a traditional secondary antibody. TidyBlot was used because it binds to the non-reduced antibody and prevents interference from denatured heavy or light chains during detection.

### Calcium uptake assay

A calcium ($Ca^{2+}$) uptake assay using Indo-1 was performed to quantify $Ca^{2+}$ uptake into the sarcoplasmic reticulum (SR) in skeletal muscle homogenates, as previously described (Geromella et al., 2023). Briefly, 50 µL of homogenized soleus or EDL was added to 300 µL of $Ca^{2+}$ uptake buffer (200 mM KCl, 20 mM HEPES, 10 mM $NaN_3$, 5 µM TPEN, 15 mM $MgCl_2$) in a 1.5 mL microcentrifuge tube. Indo-1 (I3136 MilliporeSigma) was then added (1 µL of 2 mM solution) to the microcentrifuge tube. Each mixture was plated in triplicate on an all-black 96-well plate and placed into the Molecular Devices M2 plate reader to measure the starting calcium plate. The plate was then removed, and 4 µL of ATP (250 mM) was added to each well using a repeater pipette and placed in the reader again for 30 min. A two-point calibration system was used to allow for calculation of free $Ca^{2+}$. First 10 µL of 50 mM EGTA was added allowing for a minimum measurement of $Ca^{2+}$-bound Indo-1. Second 10 µL of 5M $Ca^{2+}$ was added to saturate Indo-1 and get a maximum measurement. Using LoggerPro software (Vernier, Canada) the rates of $Ca^{2+}$ uptake were measured as the instantaneous velocity (tangent analyses) at free $Ca^{2+}$ concentrations of 250 and 500 nM. All rates were normalized to total protein as determined using BCA protein assay and were expressed as nmol of $Ca^{2+}$/g of protein/min.

### Immunofluorescent fibre typing

Immunofluorescent fibre-type staining was done on OCT-mounted soleus and EDL muscles to determine whether LiCl treatment would alter fibre-type composition. To accomplish this, we used a immunofluorescent fibre-type staining protocol as previously described (Barsanowski et al., 2022; Baranowski et al., 2023; Marcella et al., 2024). Specifically the mounted OCT muscles were cut into 10 µm-thick cryosections with a cryostat maintained at −20°C. The sections were incubated for 1 h in blocking solution (10% goat serum in 1× phosphate-buffered saline (PBS)). Primary antibodies were diluted in 10% goat serum (MHC I (BA-F8) 1:50; MHC IIa (SC-71) 1:600; MHC IIb (BF-F3) 1:100), and the sections were incubated in the primary antibody overnight. The slides were then washed thrice for 5 min each with 1× PBS. The sections were incubated in secondary antibody for 1 h in the dark (Alexa Fluor 350 IgG$_{2b}$ (blue) 1:500; Alexa Fluor 488 IgG$_1$ (green) 1:500; Alexa Fluor 555 IgM (red) 1:1000). The slides were then washed thrice for 5 min each with 1× PBS. ProLong Gold Antifade Reagent was added, and a coverslip was mounted on the slide. Slides were imaged using the BioTek Cytation 5 Multimode Plate Reader at 10× magnification using autoexposure settings for three filters: DAPI, GFP and Texas Red. Images were stitched, processed and saved using the Gen5 image processing functions. Analysis

was performed using ImageJ (NIH) software and the cell-counter tool.

## RNA sequencing

Gastrocnemius muscles from sham, OVX and OVX-Li mice ($n = 3$ per group) were homogenized in 1 mL of TRIzol Reagent (15596026, ThermoFisher Scientific, MI, USA), and RNA was extracted using the Invitrogen PureLink RNA Mini Kit (12183025, ThermoFisher Scientific). Total RNA sequencing with purified mRNA was performed by Novogene (Novogene Corporation Inc., Sacramento, CA, USA) using NovaSeqXPlus. Differentially expressed genes (DEGs) between OVX *vs*. sham, OVX-Li *vs*. OVX and OVX-Li *vs*. sham were identified (log$_2$ fold-change |log$_2$FC| > 0.25, $P < 0.05$) by Novogene using DESeq2 software, which leverages information across all genes to improve variance estimation, providing robust results even with small sample sizes (Love et al., 2014). DEGs were subsequently subjected to gene ontology (GO) pathway enrichment analysis, and select statistically significant pathways (based on adjusted $P < 0.05$ after Benjamini-Hochberg (BH) correction) were presented. Additionally the response of the top 50 DEGs between OVX and sham based on statistical significance (adjusted $P$-value after BH correction) was also depicted to illustrate the potential effects of Li treatment on the OVX muscle transcriptome. However all up- and downregulated genes, as well as significant GO pathways, are listed in Supplementary Datasets 1–5.

## Statistical analysis

The normality of all datasets was verified using the Shapiro–Wilk test before conducting statistical analysis. Comparisons involving only the group factor were analysed using a one-way ANOVA for normally distributed data or a Kruskal–Wallis non-parametric test when normality was not met (EDL: GSK3, pGSK3:GSK3 ratio, time to 90% initial force and CS content). Two-way repeated-measures ANOVAs were used to analyse force–frequency curves (main effects of group and stimulation frequency and their interaction), fatigue curves (main effects of group and time and their inter-action) and calcium uptake rates (main effects of group and free calcium concentration and their interaction). When significant effects were observed, Tukey's or Dunn's *post hoc* tests (for one-way analyses) and Fisher's least significant difference (LSD) tests (for two-way inter-actions) were applied. Statistical outliers were identified and removed using the ROUT method (Q = 1%). All analyses were performed in GraphPad Prism 10 using a significance level of $P < 0.05$. Data are presented as mean ± SD.

## Results

Uterine weight was measured at end-point to confirm successful ovariectomy surgery. As expected, uterine mass was significantly lower in OVX mice compared to sham mice for both OVX ($-75\%$) and OVX-Li ($-80\%$) mice (Table 1). There were no differences across groups for soleus and EDL muscle weights, both absolute and relative to body mass (Table 1).

Immunoprecipitation experiments were performed in the soleus and EDL muscles to identify the levels of GSK3$\beta$ inhibitory Ser9 phosphorylation as an indicator of GSK3 activation status. In the soleus, levels of immuno-precipitated GSK3$\beta$ were approximately 2.4-fold higher in OVX mice than in OVX-Li mice (Fig. 1A,B). Although no differences were detected between groups for Ser9 phosphorylated GSK3$\beta$ (Fig. 1C), when expressed relative to the total amount of GSK3$\beta$ pulled down through immunoprecipitation, we found that OVX-Li mice had trending increases in phosphorylation status compared to both sham and OVX mice (Fig. 1D). Conversely in the EDL, no differences in total immunoprecipitated GSK3$\beta$ or Ser9 phosphorylated GSK3$\beta$ were found between groups (Fig. 1E–H). Thus, we examined $\beta$-catenin content as a downstream marker of GSK3 inhibition, with significant increases in $\beta$-catenin content signifying sufficient GSK3 inhibition (Beurel et al., 2015; Huang et al., 2017; Kurgan, Bott et al., 2019; Kurgan, Whitley et al., 2019; Yang et al., 2011). As expected, OVX-Li mice exhibited approximately 1.9-fold higher $\beta$-catenin content than OVX control (Fig. 1I).

To assess the contractility of the soleus and EDL muscles, we used isometric force–frequency and fatigue protocols. In the soleus, the force–frequency curve revealed that OVX mice generated significantly less force than both sham and OVX-Li mice at submaximal and maximal stimulation frequencies, corresponding to reductions of approximately 20%–28% compared to sham and 30%–40% compared to OVX-Li mice (Fig. 2A). The twitch:tetanus ($P_t$:$P_o$) ratio serves as an indirect indicator of SR Ca$^{2+}$ release and/or myofilament calcium sensitivity. Comparisons across groups showed that OVX mice had a 20% lower $P_t$:$P_o$ ratio compared to sham mice and a 25% lower ratio compared to OVX-Li mice (Fig. 2B), indicating impaired SR calcium release and/or calcium sensitivity in OVX mice, which appeared to be restored with Li treatment. During the fatigue protocol, the soleus of OVX mice exhibited a leftward shift in the relative force-over-time curve, indicative of reduced fatigue resistance (Fig. 2C). A significant time $\times$ group interaction followed by *post*

**Table 1. Uterine weight and muscle weights in sham, OVX and OVX-Li mice (*n* = 9–10)**

|  | Sham | OVX | OVX-Li |
|---|---|---|---|
| Uterine weight (mg) | 63.8 ± 39.0 | 16.0 ± 5.3**** | 13.8 ± 3.1**** |
| Soleus weight (mg) | 8.5 ± 0.44 | 8.6 ± 0.38 | 8.0 ± 0.28 |
| Soleus:body mass (mg:g) | 0.32 ± 0.01 | 0.34 ± 0.02 | 0.30 ± 0.009 |
| EDL weight (mg) | 10.1 ± 0.01 | 9.1 ± 0.3 | 9.6 ± 0.25 |
| EDL:body mass (mg: g) | 0.38 ± 0.008 | 0.35 ± 0.01 | 0.36 ± 0.02 |

*Note*: A one-way ANOVA with Tukey's *post hoc* test was used, **** $P < 0.0001$ *vs.* Sham. All values are presented as mean ± SD.

*hoc* testing in the two-way repeated-measures model showed that differences between OVX and OVX-Li mice emerged between 50 and 70 s of the fatigue protocol – approximately the point at which force had declined by 70% (dashed line in Fig. 2C). Consistent with this, OVX-Li mice displayed a ∼1.2-fold longer time to reach 70% of the initial force compared to OVX mice (Fig. 2D); however, this difference was no longer apparent when examining the time to reach 50% of initial force (Fig. 2E). No differences were observed between the sham and OVX groups.

In the EDL, the force–frequency curve revealed that OVX mice produced lower force than both sham (−34–37%) and OVX-Li (−28–32%) mice, particularly at 40, 80 and 160 Hz (Fig. 2F). Unlike the soleus, there were no differences in the $P_t$:$P_o$ ratio among groups (Fig. 2G). During the fatigue protocol, however, a significant time × group interaction – confirmed by *post hoc* analysis in the two-way repeated-measures model – showed that differences between OVX and OVX-Li mice emerged rapidly and transiently within the first 16 s of the fatigue protocol, approximately when force had declined by 90%

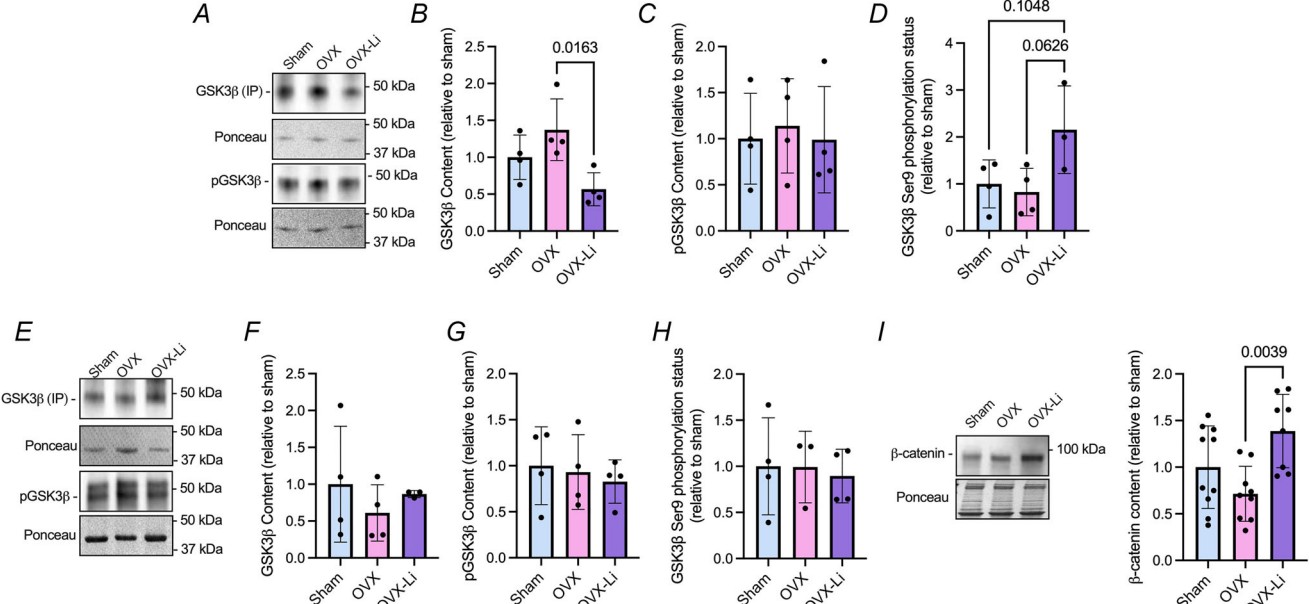

**Figure 1. Soleus and extensor digitorum longus (EDL) GSK3$\beta$ Ser9 phosphorylation status and $\beta$-catenin levels**
*A*, representative images of soleus total and phosphorylated GSK3$\beta$ content from GSK3$\beta$ immunoprecipitated eluents. *B*, total GSK3$\beta$ is greater in OVX mice compared to OVX-Li mice in the soleus. *C,D*, phosphorylated GSK3$\beta$ (Ser9) content relative to total GSK3$\beta$ content is higher in OVX-Li mice compared to OVX mice in the soleus. *E*, representative images of EDL total and phosphorylated GSK3$\beta$ content from GSK3$\beta$ immunoprecipitated eluents. *F–H*, in the EDL there are no differences between groups on total or phosphorylated GSK3$\beta$ content (*n* = 4 per group). *I*, $\beta$-catenin content is higher in the EDL from OVX-Li mice compared to sham and OVX. For all analyses a one-way ANOVA was used with a Tukey's *post hoc* test except for *F* and *H*, where a non-parametric Kruskal–Wallis test was used. For *B–D* and *F–H* *n* = 3–4 per group; for *I* *n* = 8–10 per group. All values are denoted as mean ± SD.

(dashed line in Fig. 2*H*). Consistent with this OVX-Li mice exhibited a ~1.5-fold longer time to reach 90% of the initial force compared to OVX mice (Fig. 2*I*). However this difference was no longer evident by the time force had declined to 50% of the initial value (Fig. 2*J*).

Collectively, our data show that Li supplementation enhanced specific force production and temporarily improved fatigue resistance in the soleus and EDL muscles of OVX-Li mice compared to OVX. In partial agreement with these findings, metabolic cage analysis revealed that OVX mice were 43% less mobile than sham mice (Fig. 3*A*), likely reflecting the muscle weakness and fatigue observed in Fig. 2. In contrast, this reduction in activity was attenuated in OVX-Li mice, whose cage ambulation was, on average, 19% lower than that of sham mice, and the difference was not statistically significant (Fig. 3*A*). No

differences were detected in daily energy expenditure or RER (Fig. 3*B*,*C*).

We then conducted $Ca^{2+}$ uptake assays to evaluate the enzymatic function of sarco(endo)plasmic reticulum $Ca^{2+}$ ATPase (SERCA) in the soleus and EDL. SERCA catalyses the active transport of $Ca^{2+}$ into the SR and are critical regulators of cytosolic $Ca^{2+}$ in muscle. By restoring $Ca^{2+}$ levels in the SR, SERCA not only influences muscle relaxation but also muscle contractility, ensuring a sufficient releasable $Ca^{2+}$ pool inside the SR (Periasamy & Kalyanasundaram, 2007). In the soleus, we detected a significant main effect of $Ca^{2+}$, indicating that as $[Ca^{2+}]_{free}$ increased from 250 to 500 nM the rate of $Ca^{2+}$ uptake also increased, highlighting the $Ca^{2+}$ dependency on SERCA function (Fig. 4*A*). Additionally, there was a significant main effect of group, with *post*

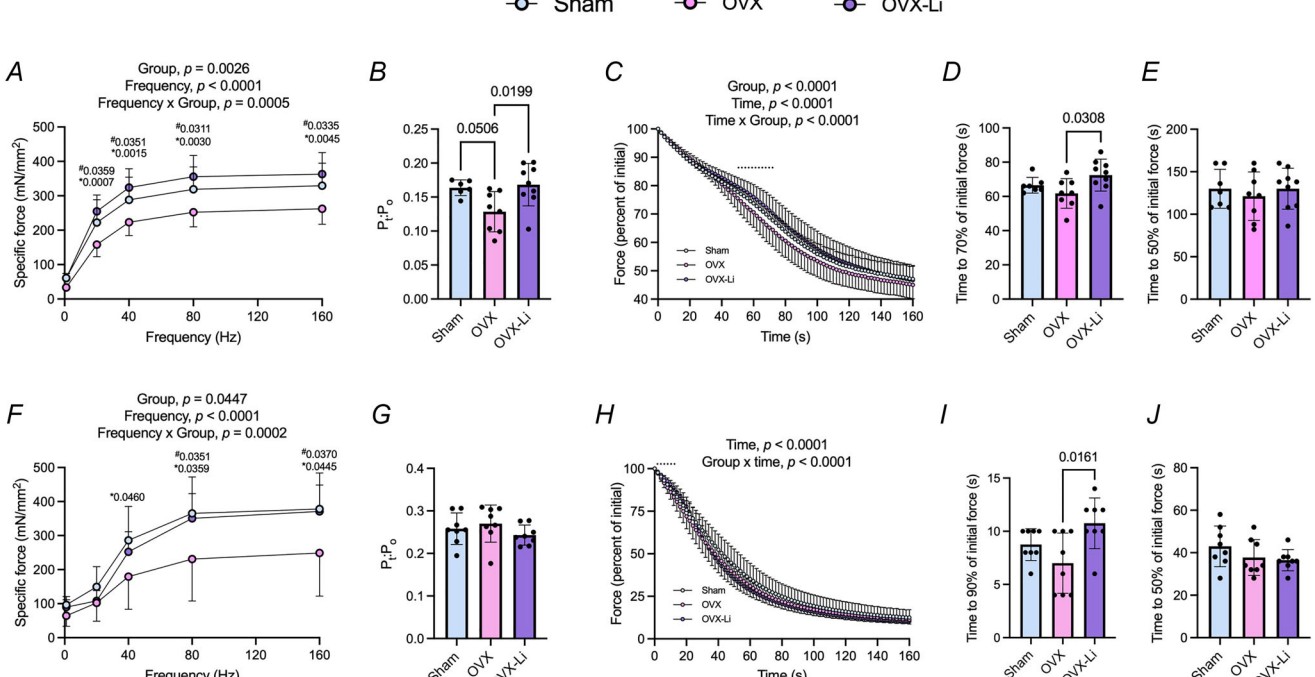

**Figure 2. Soleus and extensor digitorum longus (EDL) muscle contractility and fatigue resistance**
*A*, OVX mice have lower specific force production (mN/mm²) of the soleus than sham and OVX-Li mice. *B*, twitch: tetanus ratio ($P_t$:$P_o$) is lower in OVX soleus compared to OVX-Li and trending lower *vs.* sham. *C*, percentage of initial force plotted over time during an isometric fatigue protocol across sham, OVX and OVX-Li soleus muscles. *D*, time to reach 70% of initial force is significantly lower in the OVX soleus *vs.* OVX-Li. *E*, time to reach 50% of the initial force within the soleus is not different across groups. *F*, OVX mice have lower specific force production (mN/mm²) of the EDL than sham and OVX-Li mice. *G*, $P_t$:$P_o$ shows no difference in the EDL across groups. *H*, percentage of initial force plotted over time during an isometric fatigue protocol across sham, OVX and OVX-Li EDL muscles. *I*, time to reach 90% of initial force is significantly lower in the OVX EDL *vs.* OVX-Li. *J*, time to reach 50% of the initial force within the EDL is not different across groups. For *A* and *F* a two-way repeated ANOVA was used to assess the main effects of group and frequency and their potential interaction; *sham *vs.* OVX and #OVX *vs.* OVX-Li with exact *P*-values shown. For *C* and *H* a two-way repeated ANOVA was used to assess the main effects of group and time and their potential interaction, with a Tukey's *post hoc* test where appropriate. The dashed lines represent the range of frequencies where significant differences lie between OVX and OVX-Li muscles as determined by a *post hoc* Fisher's least significant difference (LSD) test. For *B*, *D*, *E*, *G* and *J* a one-way ANOVA was used with a Tukey's *post hoc* where appropriate. For *I* a non-parametric Kruskal–Wallis test was used alongside Dunn's multiple comparisons test. All values are denoted as mean ± SD, *n* = 9–10 per group.

*hoc* testing revealing significant least-squares (LS) mean differences between sham and OVX mice (12.37 nmol/g protein/min, $P = 0.0361$) and between OVX-Li and OVX mice (14.16 nmol/g protein/min, $P = 0.0122$). These results suggest that OVX mice exhibit lower $Ca^{2+}$ uptake compared to sham mice, regardless of $[Ca^{2+}]_{free}$, and that Li supplementation reverses this impairment (Fig. 4*A*). In the EDL, we also found significant main effects of $Ca^{2+}$ and group; however a significant interaction between the two factors indicates that OVX-Li had significantly faster rates of $Ca^{2+}$ uptake compared to sham and OVX mice but only at 500 nм $[Ca^{2+}]_{free}$ (Fig. 4*B*). Specifically, $Ca^{2+}$ uptake in OVX-Li mice was 1.6 times higher than in sham mice and 1.7 times higher than in OVX mice.

Immunofluorescent fibre-type staining and western blotting were then performed to determine whether the improvements in soleus muscle endurance found in the OVX-Li mice were due to changes in oxidative fibre/protein expression or markers of oxidative stress. We did not find any differences in fibre-type distribution

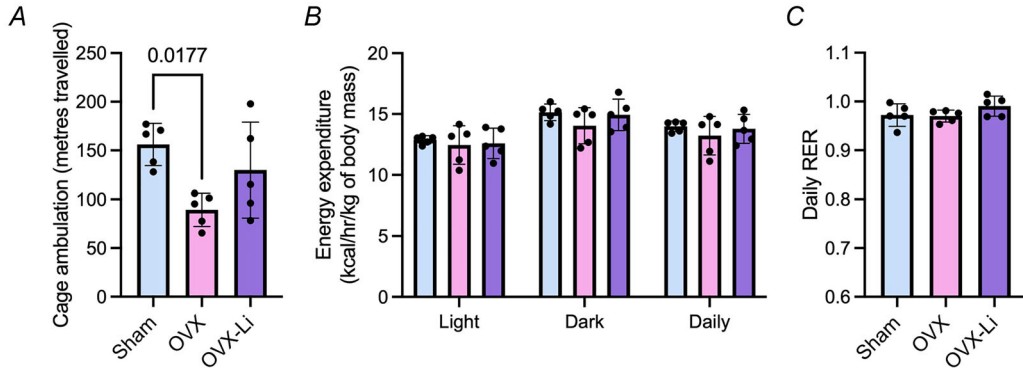

**Figure 3. Cage ambulation, daily energy expenditure and respiratory exchange ratio (RER) in sham, OVX and OVX-Li female mice**
*A*, cage ambulation is lower in OVX mice compared to sham mice. *B*, energy expenditure is not different between groups and light, dark and daily time points. *C*, RER is not different between groups. For all analysis a one-way ANOVA was used with Tukey's *post hoc* test. For B one-way ANOVAs were conducted separately for the light, dark and daily phases. All values are denoted as mean ± SD, *n* = 5 per group.

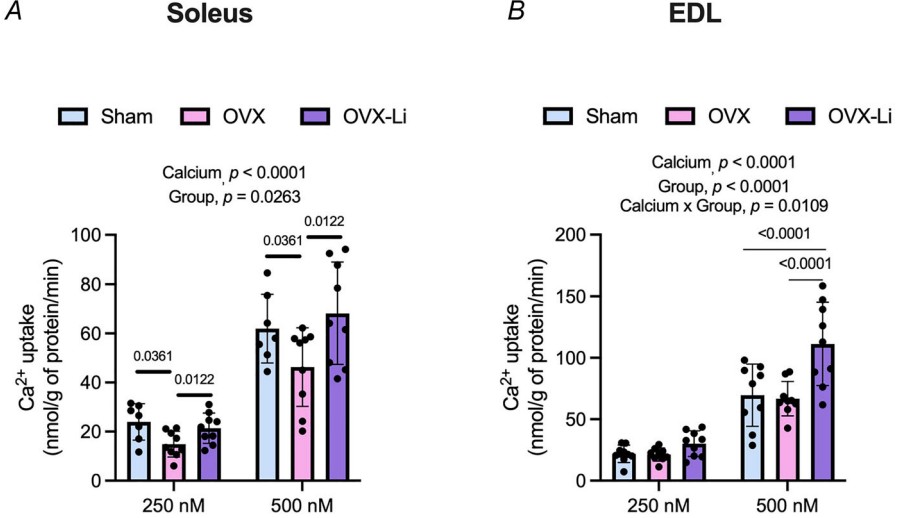

**Figure 4. SERCA-mediated $Ca^{2+}$ uptake assay in the soleus and extensor digitorum longus (EDL)**
*A*, soleus from OVX mice had significantly lower rates of $Ca^{2+}$ uptake than sham and OVX-Li mice as indicated by a significant main effect of group (*n* = 9–10 per group). *B*, EDL from OVX-Li mice had significantly higher rates of $Ca^{2+}$ uptake than sham and OVX mice at 500 nм of free $Ca^{2+}$. A two-way repeated ANOVA was used to assess main effects of free $Ca^{2+}$ concentration and group as well as their potential interaction. For *A* the main effect of group was assessed further with Fisher's least significant difference (LSD) test to determine where the differences lie within the groups. The exact *P*-values for the least-squares mean differences are shown. For *B* the significant interaction between calcium and group was assessed further with Tukey's *post hoc* test. Significant main and interactive effects are denoted in the text above. All values are denoted as mean ± SD. For *A n* = 7–9 per group; for *B n* = 9 per group.

(Fig. 5*A–E*). There were also no differences detected between groups for PGC1α content, COXIV content and CS content (Fig. 5*F–I*). However, soleus muscles from OVX mice exhibited significantly greater 4-HNE content compared to sham (1.4-fold higher, Fig. 5*J*). This effect was blunted in the OVX-Li group, where 4-HNE levels were only 1.1-fold higher than those in the sham group and were not statistically significant (Fig. 5*J*). Moreover, compared to OVX mice, OVX-Li mice showed approximately 17% lower 4-HNE levels, although this difference only approached statistical significance

($P = 0.0640$). Collectively, these findings suggest that the OVX soleus exhibits increased oxidative stress, and that Li supplementation can modestly reduce this effect (Fig. 5*J*).

In the EDL, we again found no differences in fibre-type composition across sham, OVX and OVX-Li groups (Fig. 6*A–E*). Additionally no differences were detected across groups for PGC1α or COXIV content (Fig. 6*F–H*); however, OVX mice had significantly lower CS content compared to sham (−33%) and OVX-Li mice (−18%), indicating that certain mitochondrial markers are reduced in OVX mice and partly restored with Li supplementation

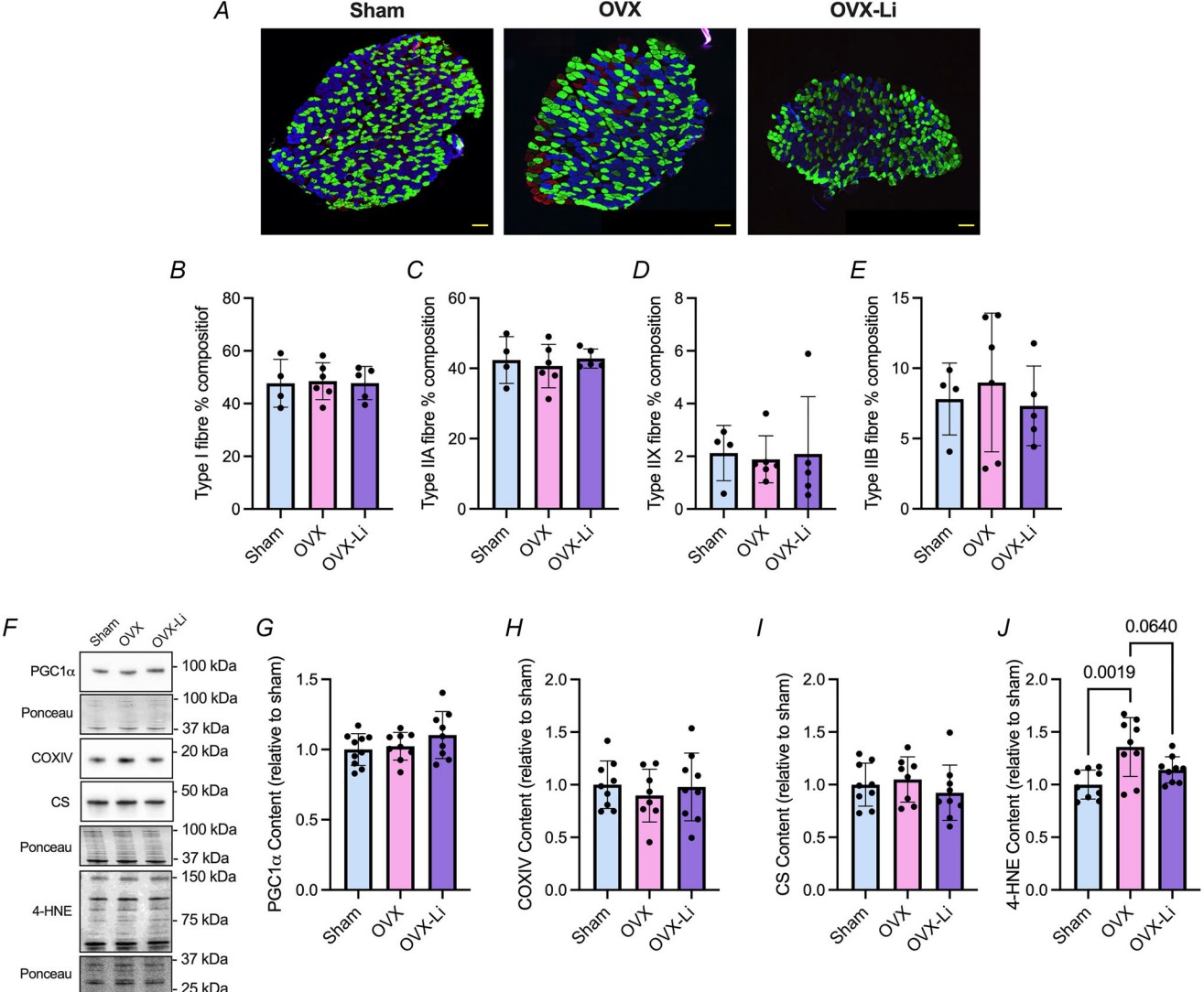

**Figure 5. Immunofluorescent fibre-type analysis and western blot analysis of markers related to oxidative metabolism and oxidative stress in the soleus**
*A*, representative immunofluorescent fibre-type staining images in the soleus (scale bars set to 1000 μm). *B–E*, percentage distribution of fibre types I, IIA, IIX and IIB. *F*, representative images of PGC1α, COXIV, CS and 4-HNE. *G*, PGC1α content is not different between groups. *H*, COXIV content is not different between groups. *I*, CS content is not different between groups. *J*, 4-HNE content is elevated in OVX mice compared to sham, but this effect is blunted in OVX-Li mice. COXIV and CS were run on the same gel and therefore have the same Ponceau. For *B–E* and *G–J* a one-way ANOVA was used. All values are denoted as mean ± SD. For *B–E* n = 4–6 per group; for G–J n = 8–10 per group.

(Fig. 6*I*). There were no differences in 4-HNE content across groups.

To further investigate the potential mechanisms underlying the improvements in specific force production and fatigue resistance, RNA was extracted from gastrocnemius muscles for RNA-Seq analysis performed by Novogene (Sacramento, USA). Across the targeted comparisons of OVX *vs.* sham, OVX-Li *vs.* OVX and OVX-Li *vs.* sham a range of 1057–1525 DEGs were identified, including both unique and overlapping genes among groups (Fig. 7*A–D*). GO enrichment analysis of the DEGs between OVX and sham mice revealed significant

upregulation in pathways related to the adaptive immune response and the proliferation of inflammatory cells (Fig. 7*E*). In contrast, pathways linked to muscle tissue development were downregulated (Fig. 7*E*). These effects were reversed with Li treatment, as evidenced by GO enrichment analysis, which demonstrated significant downregulation of pathways associated with the adaptive immune response and proliferation of inflammatory cells, alongside a significant upregulation of pathways linked to muscle tissue development and contractile components (Fig. 7*F*). Furthermore, GO enrichment pathways associated with muscle relaxation, SR calcium

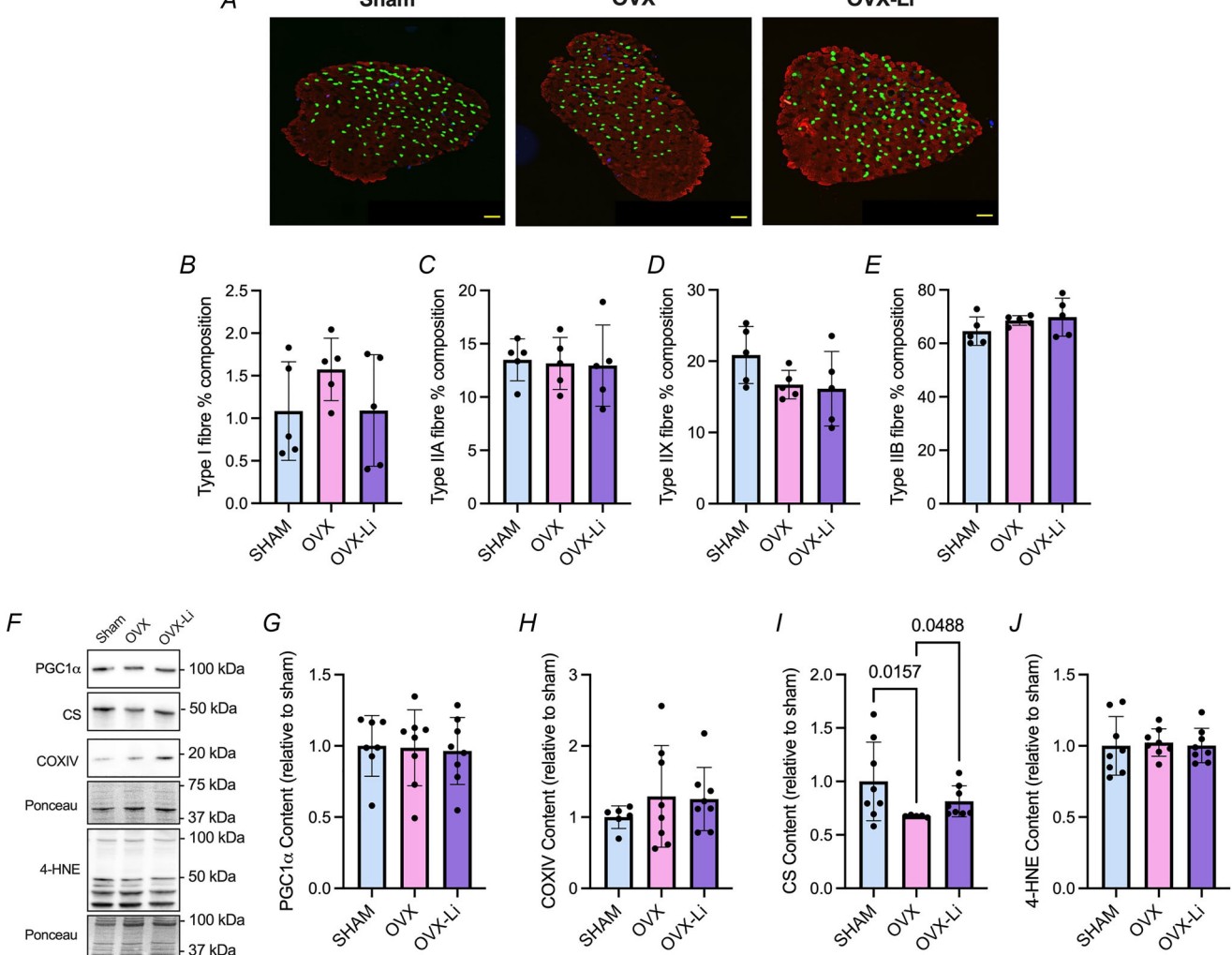

**Figure 6. Immunofluorescent fibre-type analysis and western blot analysis of markers related to oxidative metabolism and oxidative stress in the extensor digitorum longus (EDL)**
*A*, representative immunofluorescent fibre-type staining images in the EDL (scale bars set to 1000 μm). *B–E*, percentage distribution of fibre types I, IIA, IIX and IIB. *F*, representative images of PGC1α, COXIV, CS and 4-HNE. *G*, PGC1α content is not different between groups. *H*, COXIV content is not different between groups. *I*, CS content is significantly lower in the OVX EDL compared to sham and OVX-Li. *J*, 4-HNE content is not different between groups. PGC1α, COXIV and CS were run on the same gel and therefore have the same Ponceau. For *B–E*, *G–H* and *J* a one-way ANOVA was used. For *I* a non-parametric Kruskal–Wallis test was used alongside Dunn's multiple comparisons test. All values are denoted as mean ± SD. For *B–E* $n$ = 4–6 per group; for *G–J* $n$ = 5–8 per group.

transport, calcium/calcineurin-NFAT signalling and slow fibre-type switching were also significantly upregulated in OVX-Li mice compared to OVX mice (Fig. 7*F*). With respect to SR calcium handling, significant upregulations in *Atp2a2*, *Casq1* and *Casq2*, which encode for SERCA2, calsequesterin 1 and 2, respectively, were significantly upregulated in OVX-Li mice *vs.* OVX (Fig. 7*B*). These results are consistent with the observed improvements in $Ca^{2+}$ uptake. In fact similar increases in *Atp2a2* and *Casq2* were observed in OVX-Li mice *vs.* sham (Fig. 7*C*) mice, which can also potentially contribute to the greater $Ca^{2+}$ uptake found in the OVX-Li EDL *vs.* sham.

For the transition towards a slow fibre type (Fig. 7*F*), *Tnni1*, *Tnnt1* and *Tnnc1* – which encode the components of the slow troponin complex – were all significantly upregulated in OVX-Li *vs.* OVX. This was accompanied by increased expression of *Myh7* and *Myh2* (Fig. 7*B*), which encode myosin heavy chains I and IIa, respectively, both characteristic of oxidative fibres. Combined with the upregulation of *Atp2a2*, which is also highly expressed in slow oxidative fibres, these data suggest that Li supplementation in OVX mice, at least at the transcriptional level, promotes genetic reprogramming towards the oxidative fibre type. However, this did not trans-

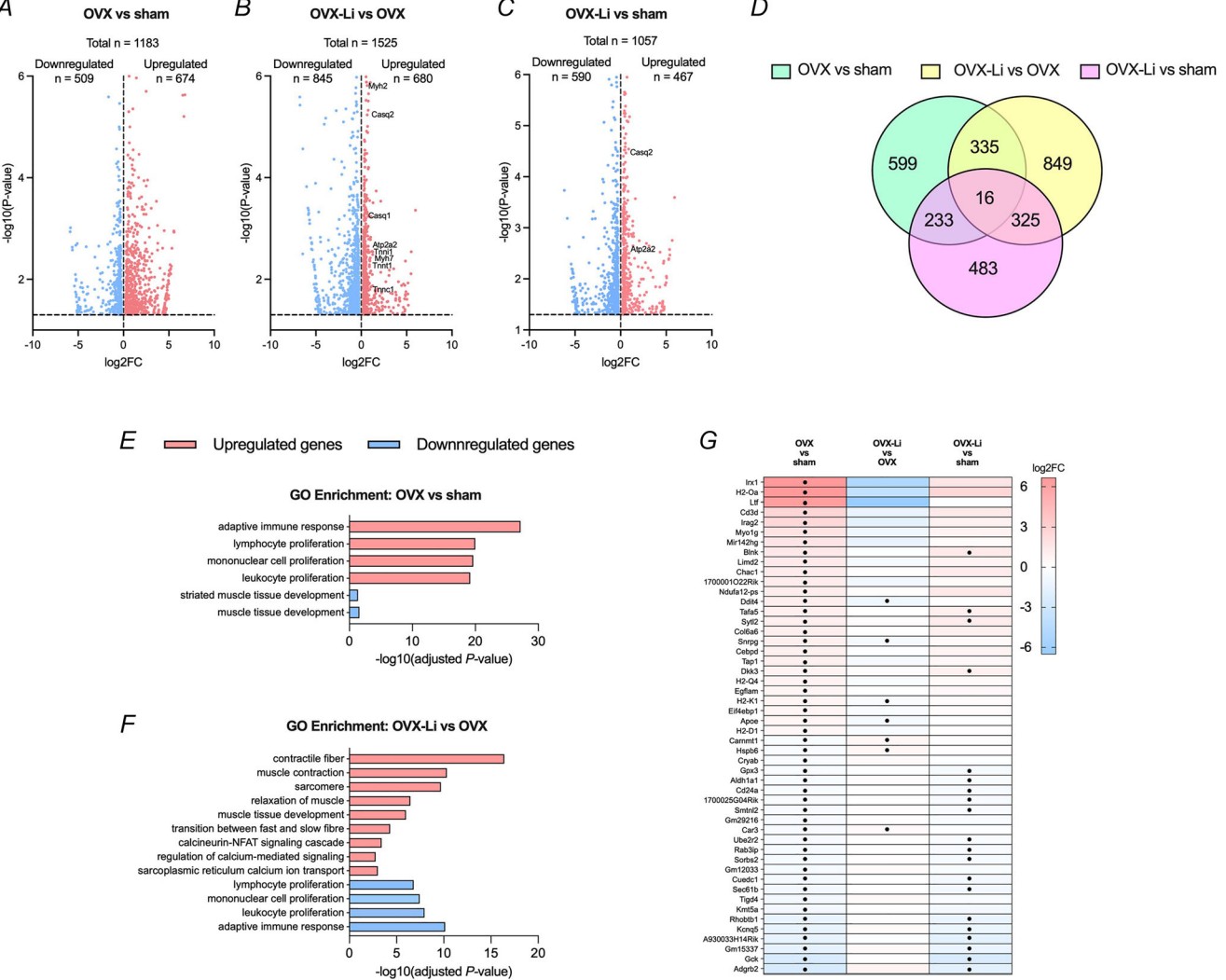

**Figure 7. RNA-Seq analysis in gastrocnemius muscles from sham, OVX and OVX-Li mice (*n* = 3 per group)**
*A–C*, volcano plots of differentially expressed genes (DEGs) identified from targeted comparisons between OVX and sham; OVX-Li and OVX; and OVX-Li and sham. The dashed line at $y$ = 1.3 indicates the significance threshold ($-\log10(p)$ = 0.05; $P$ < 0.05). Targeted genes related to GO Enrichment Pathways: sarcoplasmic reticulum (SR) $Ca^{2+}$ handling and fast-to-slow fibre transitions are depicted in *B* and *C*. *D*, Venn diagram of DEGs for OVX *vs.* sham, OVX-Li *vs.* OVX and OVX-Li *vs.* sham comparisons. *E,F*, Selected top significant GO Enrichment terms for OVX *vs.* sham and OVX-Li *vs.* OVX comparisons. *G*, top 50 DEGs based on statistical significance (adjusted $P$ < 0.05) between OVX and sham mice, with red indicating upregulated and blue indicating downregulated genes. log2FC, log2 fold-change. Exact $P$-values for these genes can be found in Supplementary Tables 3–5.

late into detectable changes in fibre-type composition as assessed through immunofluorescent analysis of the myosin heavy chain isoforms (Figs 5 and 6).

Additional exploratory analysis involving heatmaps of the top 50 DEGs, based on statistical significance with an adjusted $P<0.05$, in the OVX mice *vs.* sham displayed an inverse expression pattern following Li supplementation (Fig. 6*E*). Although this normalizing effect of Li was not significant for all genes, the resulting shift in expression indicated that the OVX-Li profile was, for the most part (30 out of 50 genes), not statistically different from sham, particularly among the upregulated genes. Overall, the RNA-Seq data suggest that Li supplementation alleviates the effects of OVX surgery on the muscle transcriptome, supports the observed improvements in muscle contractility and calcium handling and offers additional mechanistic insights through decreases in inflammatory processes and increases in muscle tissue development.

Aside from the muscle outcome measures, we next examined whether Li supplementation could restore BMD in OVX mice. An iNSiGHT Small Animal DXA scanner was used to measure BMD and body composition. As expected BMD was 8% lower in OVX mice compared to sham mice (Fig. 8*A*). However, this reduction was prevented by Li supplementation, as BMD in OVX-Li mice was 6% greater than in OVX mice and comparable to sham mice (Fig. 8*A*). No significant differences were detected in body mass, % fat mass or % lean mass (Fig. 8*B–D*).

Finally, GTT and ITT were performed to evaluate glucose regulation and insulin sensitivity. In the GTT, a significant time × group interaction indicated that both OVX and OVX-Li mice exhibited impaired glucose tolerance, with blood glucose levels remaining significantly higher throughout the test and area under the curve (AUC) values significantly greater than those of sham mice (1.5-fold higher in OVX and 1.4-fold higher in OVX-Li; Fig. 9*A*,*B*). During the ITT, another significant time × group interaction was observed; however, only the OVX mice showed a notable impairment in insulin tolerance, with a 14% greater AUC than the sham (Fig. 9*C*,*D*). Conversely, the ITT AUC was 10% higher in the OVX-Li group *vs.* sham, but this difference was not statistically significant, indicating a modest improvement in insulin sensitivity with Li supplementation (Fig. 9*C*,*D*).

## Discussion

In the present study, we examined whether low-dose Li supplementation could mitigate musculoskeletal and metabolic impairments in OVX mice. Overall, Li treatment yielded several beneficial effects, including increased specific force generation in both soleus and EDL muscles, temporary improvements in fatigue resistance, enhanced SERCA-mediated $Ca^{2+}$ uptake and increased BMD – with statistically significant differences being observed between OVX-Li and OVX mice. Other benefits, namely, improved insulin tolerance and cage ambulation, were seen in OVX-Li mice. Although not statistically different from OVX mice, they were also not significantly different from sham mice, indicating partial restoration towards sham control. Altogether, these findings in the OVX mouse model support the therapeutic potential of Li as a protective intervention against the deleterious effects of menopause. However, further studies, as outlined below, are necessary to clarify the underlying mechanisms and to assess the sustainability and translational relevance of these Li-induced benefits across different models.

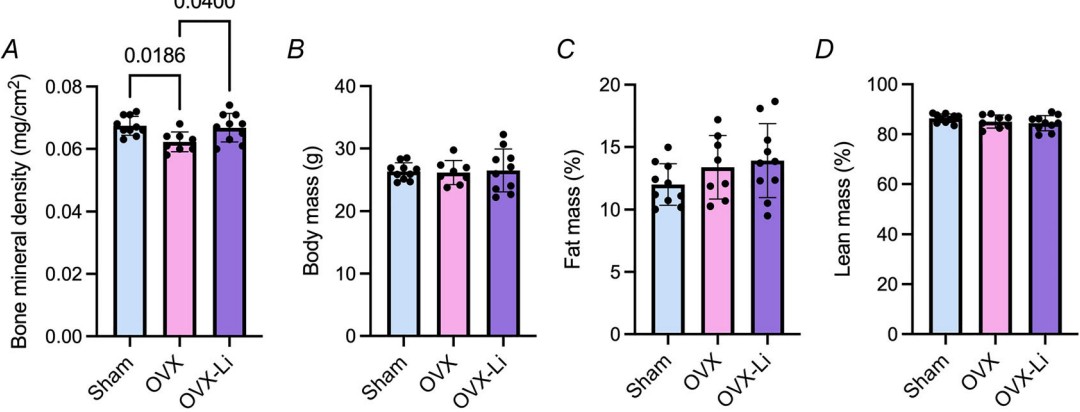

**Figure 8. Bone mineral density (BMD), body mass and percentage fat and lean mass in sham, OVX and OVX-Li mice**

*A*, OVX mice have significantly lower BMD (mg/cm²) compared to sham and OVX-Li mice. *B–D*, no differences were found between groups for total body weight (g), lean mass relative to total body weight (%) and fat mass relative to total body weight (%). For all analyses a one-way ANOVA was used with Tukey's *post hoc* test. All values are denoted as mean ± SD, *n* = 8–10 per group.

To test the effects of GSK3 inhibition with Li, we treated 28-week-old OVX mice with 50 mg/kg/day of LiCl for 8 weeks through their drinking water. Our lab has shown that a dose of 50 mg/kg/day equates to a serum Li concentration of 0.15 ± 0.05 mM (Marcella et al., 2024), which is well below the therapeutic range of Li (0.5–1.2 mM) typically used for bipolar disorder (Hamstra et al., 2023). A dose below the typical range used in the clinical setting for humans was chosen because doses within this range can still result in kidney toxicity when given to mice (Nespital et al., 2021). In the soleus, total GSK3β content was lower in OVX-Li mice than OVX mice, and OVX-Li mice had a trending greater ratio of phosphorylated GSK3β (Ser9) content relative to total GSK3β content than OVX mice. Conversely, in the EDL there were no differences detected between groups for total GSK3β content or the ratio of phosphorylated GSK3β (Ser9) content relative to total GSK3β content. Nonetheless, this does not discount GSK3 inhibition as Li can regulate GSK3 in two ways: a secondary and indirect path that relies on the activation of kinases (e.g. Akt, PKA, PKC), which phosphorylate and inhibit GSK3; and a primary and direct path where Li competes with $Mg^{2+}$ (a cofactor for GSK3 activity) for a binding site on GSK3 (Jope, 2003). Corresponding well with this, we found an increase in β-catenin content with Li supplementation compared to OVX mice. β-catenin is part of the canonical Wnt signalling pathway, and its phosphorylation by GSK3 signals for its cellular degradation; thus an increase in β-catenin content can be used as a downstream marker of GSK3 inhibition that also appears to act independently of Ser9 GSK3 phosphorylation (Beurel et al., 2015; Huang et al., 2017; Kurgan, Bott et al., 2019; Kurgan, Whitley et al., 2019; Yang et al., 2011).

To assess the functional outcomes associated with Li supplementation in skeletal muscle, we measured soleus and EDL specific force production and fatigue resistance using *ex vivo* muscle contractile experiments. Consistent with previous studies, we found that the soleus and EDL of OVX mice produced significantly less force than sham mice (Moran et al., 2006; Moran et al., 2007). Li supplementation in OVX mice restored the force-generating capacity of both muscles, as indicated by force–frequency curves that were significantly greater than those of OVX controls and comparable to sham. This force-stimulating effect is consistent with our previous findings in male C57BL/6J mice (Whitley et al., 2020). With respect to fatigue, our results show that Li supplementation transiently enhances fatigue resistance in both the OVX-Li soleus and EDL muscles compared to OVX. This effect occurred within the first 50–70 s of the fatigue protocol in the soleus and within the first 16 s in the EDL, extending the time required to reach 70% and 90% of the initial force, respectively. However this benefit was no longer apparent once the force declined to 50% of its initial value, perhaps due to the saturating nature of the fatigue protocol used in this study. Furthermore, using the Promethion metabolic caging system we found that OVX mice exhibited a 43% reduction in cage ambulation compared to sham, whereas OVX-Li mice showed only a 19% reduction, which was not statistically significant. We interpret this as a partial restoration of cage ambulation, potentially linked to the improved force-generating capacity observed at the individual muscle level.

In the soleus, the improvements in specific force production may be driven by a frequency-dependent increase in $Ca^{2+}$ sensitivity and/or $Ca^{2+}$ release, enhanced SERCA function and reduced oxidative stress, as indicated by lower 4-HNE content – an end-product of lipid peroxidation (Shoeb et al., 2014) – compared to the sham. A higher $P_t:P_o$ ratio reflects greater SR $Ca^{2+}$ release

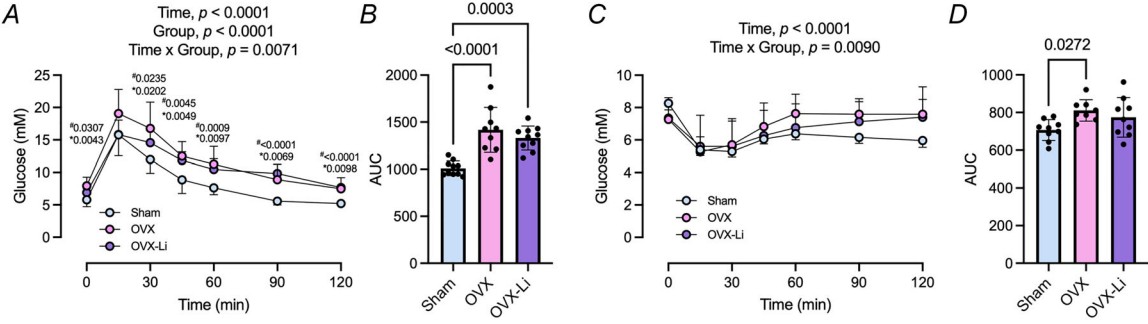

**Figure 9. Glucose handling in sham and OVX female mice**
*A,B*, glucose tolerance tests and the area-under-the-curve (AUC) analysis show that both OVX and OVX-Li mice have impaired glucose tolerance compared to sham. *C,D*, insulin tolerance tests and AUC analysis demonstrate that only OVX mice, not OVX-Li, have impaired insulin tolerance compared to sham. For *A* and *C* a two-way repeated ANOVA was used to assess main effects of group and time, with Tukey's *post hoc* test being used if a significant interaction was detected. For *B* and *C* a one-way ANOVA with Tukey's *post hoc* test was used. All values are denoted as mean ± SD, *n* = 9–10 per group. For *A* *sham *vs*. OVX, and #sham *vs*. OVX-Li with exact *P*-values shown.

and/or increased calcium sensitivity of the myofilaments, as maximal tetanic force induces maximal SR $Ca^{2+}$ release and full occupancy of calcium-binding sites on troponin. OVX mice exhibited a lower $P_t$:$P_o$ ratio compared to both sham (approaching statistical significance) and OVX-Li (significantly different), indicating impairments in SR $Ca^{2+}$ release and/or myofilament $Ca^{2+}$ sensitivity that were restored with Li supplementation. Moreover the increase in $P_t$:$P_o$ ratio in the OVX-Li soleus may also suggest enhanced neuromuscular efficiency, whereby a lower stimulation frequency is needed to achieve the same force output (Laidlaw & Vandenboom, 2022). With respect to SERCA, SERCA-mediated $Ca^{2+}$ uptake was impaired in the OVX soleus compared to sham but restored in the OVX-Li group. Together, these improvements in SR $Ca^{2+}$ handling, release, myofilament $Ca^{2+}$ sensitivity and neuromuscular efficiency likely contribute to the enhanced specific force production observed in the soleus of OVX-Li mice.

For oxidative stress, 4-HNE was selected as a biological marker because oestrogen has been found to function similarly to cholesterol and stabilize cells membranes, thereby protecting muscle cell membranes from lipid peroxidation (Pellegrino et al., 2022). Thus, as anticipated with the absence of oestrogen, 4-HNE levels were 1.4-fold higher in soleus muscles from OVX mice compared to sham mice. In contrast, 4-HNE levels in the OVX-Li soleus were only 1.1-fold higher *vs.* sham mice, and this was not statistically significant. Moreover, when compared to the OVX soleus, 4-HNE levels tended to be 17% lower in the OVX-Li soleus, and although this did not quite reach statistical significance, these data suggest that Li supplementation can partly limit the oxidative stress, as indicated by 4-HNE content, which could also contribute to the improvements in soleus muscle-specific force production. However, we acknowledge that our study is limited to only using one marker of oxidative stress, and other measures of muscle quality should be explored in the future.

The transient improvement in fatigue resistance observed in the OVX-Li soleus prompted us to examine potential underlying mechanisms, including fibre-type distribution and markers of mitochondrial content. However, Li supplementation did not alter fibre-type distribution or increase the protein levels of select mitochondrial markers, as no differences were detected in PGC1$\alpha$, COXIV or CS. This contrasts with our previous findings in male mice (Whitley et al., 2020), suggesting that some effects of Li supplementation may be sex-dependent. Given the lack of changes in fibre type or mitochondrial content the enhanced fatigue resistance in the OVX-Li soleus is more likely attributable to improved SR $Ca^{2+}$ handling and reduced oxidative stress. Enhancements to SERCA function and lower 4-HNE levels would be expected to support greater $Ca^{2+}$ reuptake

and availability within the SR during repeated contractions. Because fatigue is associated with diminished SR $Ca^{2+}$ release, partly due to SERCA dysfunction (Nogueira et al., 2013; Sharlo et al., 2023; Tupling, 2004), the improved SERCA-mediated $Ca^{2+}$ uptake observed with Li treatment would help maintain sufficient releasable $Ca^{2+}$, thereby contributing to the observed improvements in soleus endurance. Furthermore the greater $P_t$:$P_o$ ratio found in OVX-Li soleus muscles *vs.* OVX mice could also signify enhanced myofilament $Ca^{2+}$ sensitivity, which, itself, could also improve fatigue resistance.

Unlike the soleus, there were no differences in the $P_t$:$P_o$ ratio among groups within the EDL, suggesting that neither the improvements in specific force production nor the enhanced fatigue resistance observed in the OVX-Li EDL compared to the OVX EDL can be attributed to differences in SR $Ca^{2+}$ release and/or sensitivity. Yet SERCA-mediated $Ca^{2+}$ uptake was significantly greater in the OVX-Li EDL compared to both OVX and sham mice, which may have contributed to the observed improvements in specific force production and fatigue resistance. The finding that SERCA function was enhanced beyond that found in sham mice, despite no additional gains in force or fatigue resistance relative to sham, likely reflects a reduction in muscle quality induced by the OVX condition.

Within the EDL, there were no differences in 4-HNE content. Although this is only one marker of oxidative stress, these results suggest that unlike the soleus, the OVX EDL may not experience oxidative stress, possibly due to its lower activity pattern. The soleus is a postural muscle that exhibits a more tonic activity pattern throughout the day, whereas the EDL exhibits a more phasic activity pattern (Schiaffino & Reggiani, 2011). Therefore the absence of oestrogen, combined with greater activation throughout the day, would likely lead to increased oxidative stress in the OVX soleus but not necessarily in the EDL. Further in line with muscle-type-dependent changes in response to OVX surgery, CS within the EDL was significantly lower in the OVX mice compared to sham, but were restored to sham levels in the OVX-Li group. This indicates that for this specific mitochondrial content marker, Li supplementation rescued its downregulation. Although other markers of mitochondrial biogenesis signalling and content, such as PGC1$\alpha$ and COXIV, respectively, were not different across groups, these data still suggest that the EDL might have improved mitochondrial function, which could contribute to the improvements in both specific force and fatigue resistance within the EDL; however future experiments focused on examining mitochondrial function in the EDL (and the soleus) are required.

Another possible contributing factor to the transient improvements in fatigue resistance with Li treatment in both the EDL and soleus could be increased muscle

glycogen content. Glycogen depletion is tightly connected to fatigue (Nielsen et al., 2014; Ørtenblad et al., 2013); and GSK3 is well known to negatively regulate glycogen synthase activity and thus glycogen synthesis (Hockey et al., 2025). It is also worth noting, particularly for the EDL, that the transient improvements in fatigue resistance observed in the OVX-Li group occurred very early, only within the first 16 s of stimulation. This might suggest that Li supplementation may have altered critical muscle metabolites. Specifically, we speculate that Li may increase phosphocreatine availability within the EDL, and that once these stores are depleted any benefit to fatigue resistance is lost. Therefore, future research should explore the possible effects of Li supplementation on muscle metabolites, such as phosphocreatine and glycogen, in the OVX muscle.

In an attempt to gain further mechanistic insight into how Li supplementation benefits muscle function in OVX mice, RNA-Seq was performed commercially (Novogene) on total RNA extracted from gastrocnemius muscles. The results largely suggest that Li supplementation attenuates and possibly even reverses the effects of OVX surgery on the muscle transcriptome. Compared to the sham group, OVX surgery led to a significant upregulation in GO terms related to inflammation and a significant downregulation in terms related to muscle tissue development. However, these effects were essentially absent when OVX mice were treated with Li, as OVX-Li mice showed significant upregulation in GO terms associated with muscle tissue development and significant downregulation in GO terms linked to inflammation when compared to OVX mice.

Furthermore, OVX-Li mice exhibited significant upregulations in GO terms related to SR calcium handling and calcium signalling, supporting some of our findings presented in the current study. Specifically *Atp2a2*, *Casq1* and *Casq2* genes were all significantly upregulated in the OVX-Li mice *vs*. OVX (Fig. 7*B*). These genes encode SERCA2a, calsequestrin 1 and calsequestrin 2, respectively, and may help explain the increased $Ca^{2+}$ uptake observed in both the OVX-Li soleus and EDL muscles compared to OVX. An increase in SERCA2a content indicates that more SERCA pumps are available to catalyse the transport of $Ca^{2+}$; and an increase in the SR calcium-binding protein, calsequesterin, would help limit SR $[Ca^{2+}]_{free}$, thereby driving $Ca^{2+}$ uptake forward and into the SR (Tupling, 2004). These results are partly consistent with our previous reports showing that Li supplementation can increase SERCA2 content in the male murine heart, though calsequesterin content was not measured (Hamstra et al., 2020). Moreover, the fact that similar increases in *Atp2a2* and *Casq2* genes were found in OVX-Li mice *vs*. sham (Fig. 7*C*) aligns with the greater $Ca^{2+}$ uptake found in the OVX-Li EDL relative to sham.

With respect to the increased $P_t$:$P_o$ ratio found in the OVX-Li soleus compared to OVX – which suggests a potential enhancement in myofilament $Ca^{2+}$ sensitivity – the RNA-Seq data show that *Tnni1*, *Tnnt1* and *Tnnc1*, the genes encoding components of the slow troponin complex (slow troponin I, T and C, respectively), were among the significantly upregulated DEGs in OVX-Li mice. This is noteworthy because the slow troponin complex is more calcium-sensitive than its fast-troponin counterpart (Rasmussen & Jin, 2021), consistent with the functional changes we observed in the OVX-Li soleus. In contrast, no differences in $P_t$:$P_o$ were detected in the EDL across groups, which highlights muscle-type-dependent effects of Li and underscores a key limitation of the RNA-Seq data; the sequencing was not performed on isolated soleus and EDL muscles, making muscle-specific inter-pretation difficult. In any case more targeted experiments using permeabilized fibres from both muscles following Li supplementation are required to directly determine whether $Ca^{2+}$ sensitivity is altered and whether this effect is differentially affected by muscle types.

Another apparent disconnect between the RNA-Seq data and the physiological findings from the soleus and EDL pertains to the immunofluorescent fibre-type analyses. RNA-Seq data from the gastrocnemius muscles revealed a significant upregulation in the GO enrichment pathway related to the transition between fast and slow fibres. Specifically, genes associated with the slow/oxidative fibre type, *Atp2a2*, *Tnni1*, *Tnnt1*, *Tnnc1*, *Myh7* and *Myh2*, were all significantly upregulated in OVX-Li *vs*. OVX. This aligns well with current evidence showing that GSK3 inhibition can promote the oxidative fibre type, in part, through the activation of calcineurin signalling (Hockey et al., 2025), which was also identified as a significantly upregulated GO pathway in Fig. 7*F*. However, this transcriptional change did not translate into measurable differences in fibre-type composition as assessed by immunofluorescence microscopy using myosin heavy chain isoform-specific antibodies in either soleus or EDL. In any case, the upregulation of these genes in OVX-Li muscle suggests that Li supplementation triggers a genetic reprogramming toward a slow-oxidative phenotype, further warranting future studies that directly measure the oxidative potential of the OVX muscle in response to Li supplementation (i.e. mitochondrial respiration). Indeed, we cannot rule out the possibility that other proteins associated with the oxidative phenotype, which were not measured in this study, may have also contributed to the improvements in fatigue resistance.

A final exploratory approach used to assess the potential effects of Li supplementation on the muscle transcriptome was to determine whether it would mitigate the genetic reprogramming caused by OVX surgery. To this end, we examined what the response of the top 50 DEGs

found between OVX mice *vs.* sham mice might be after Li supplementation. Strikingly, we found an inverse expression pattern in the OVX-Li *vs.* OVX comparison, suggesting that Li supplementation attenuates the effects of OVX surgery on the muscle transcriptome. One notable gene is *Col6a6*, which was significantly upregulated in OVX muscle compared to sham, but then downregulated after Li supplementation to levels no longer statistically different from sham mice. This gene encodes for the protein, collagen VI, which is known to be linked to fibrosis in a wide range of organs (Williams et al., 2022). In muscle, increased fibrosis can contribute to muscle weakness, as observed in cases of muscular dystrophy (Mogharehabed & Czubryt, 2023). Therefore a reduction in fibrosis may contribute to the improvement in muscle strength observed in OVX-Li mice. Another noteworthy gene is *Hspb6*, which encodes for the protein heat shock protein B6 (HSBP6), a 17 kDa protein that belongs to the small heat shock protein family (Mogharehabed & Czubryt, 2023). Though the role of HSBP6 in skeletal muscle is largely understudied (Dreiza et al., 2010), it has been shown to stimulate cardiac muscle contractility partly by increasing $Ca^{2+}$ uptake (Pipkin et al., 2003). Therefore, Li supplementation could potentially enhance SERCA function and, ultimately, muscle strength and endurance in skeletal muscle by increasing HSBP6, although further studies are necessary to confirm this.

In addition to muscle fatigue and weakness, osteoporosis is highly prevalent among postmenopausal females and is characterized by low BMD and deterioration of bone structure that results in an increased risk of frailty and fractures (Ji & Yu, 2015). Postmenopausal females have a four times higher rate of osteoporosis compared to males, and oestrogen deficiency is a main contributing factor (Alswat, 2017). Oestrogen plays a key role in regulating bone turnover by inducing formation and inhibiting bone resorption, mediated by osteoblast cells and osteoclast cells, respectively (Khosla et al., 2012). Although some bone loss is inevitable with ageing, the natural cessation of oestrogen during menopause leads to an uncoupling of these bone cells, accelerating the rate of bone resorption. In turn, a rapid net loss of bone mass and unfavourable changes in bone structure ultimately lead to an increased risk of fragility fracture (Bjørnerem et al., 2018; Greendale et al., 2020; Ji & Yu, 2015; Karlamangla et al., 2018). In the present study, we found that BMD was significantly reduced in OVX mice compared to sham mice, which is consistent with previous studies (Felson et al., 1993). This phenotype, however, was prevented by low-dose Li supplementation, as OVX-Li mice also showed significantly higher BMD levels than OVX mice, and these levels were comparable to sham. This is not entirely surprising as GSK3 has been implicated in osteoporosis

as it inhibits the anabolic Wnt/$\beta$-catenin pathway (Bai et al., 2019; Marsell et al., 2012; Vachhani et al., 2018). In turn, GSK3 inhibition has been shown to increase bone formation, healing and strength (Bai et al., 2019; Kurgan, Bott et al., 2019; Vachhani et al., 2018). We have previously shown that treating WT male mice with an even lower dose of Li (10 mg/kg/day) effectively inhibited GSK3$\beta$ in bone, which resulted in enhanced Wnt/$\beta$-catenin signalling (Kurgan et al., 2019b). Additionally, a previous study found that OVX rats treated with Li (serum Li concentrations not disclosed) had significant improvements in BMD and bone structure (Bai et al., 2019), which altogether supports the notion that Li supplementation may be able to promote musculoskeletal health in females undergoing menopause.

With respect to metabolism, we also assessed glucose handling given that postmenopausal females and OVX mice have lowered insulin sensitivity and impaired glucose regulation (Bermingham et al., 2022; Cooper et al., 2007; Hayward et al., 2021; Zidon et al., 2020). Similarly, we found that OVX mice displayed impairments to both glucose and insulin tolerance when compared to sham mice. However, although OVX-Li mice appeared to be glucose intolerant, their insulin tolerance was not significantly different from that of sham mice. Overall, these findings indicate that low-dose Li supplementation led to an enhancement in insulin sensitivity among OVX mice. Although glucose tolerance did not show improvement in OVX-Li mice, it is possible that the insulin levels needed to trigger a response during the glucose tolerance test were reduced in these mice. This phenomenon has been noted in transgenic mice engineered to overexpress GSK3; despite both wild-type and GSK3-overexpressing mice exhibiting similar glucose tolerance at 10 weeks of age the GSK3-overexpressing group secreted more insulin during the GTT to achieve the same AUC (Pearce et al., 2004). In fact, GSK3 is a well-known target for diabetes, and Li is known to exert insulin-mimetic effects and enhance glucose transport in skeletal muscle (Macko et al., 2008; Marcella, Copeland et al., 2022; Rodriguez-Gil et al., 1993; Rossetti, 1989), and thus, the finding that Li improved insulin sensitivity, albeit modestly, is still consistent with previous reports.

Collectively, our data show that OVX-Li mice experienced different levels of protection from Li in terms of muscle strength, endurance, BMD and insulin sensitivity. The improvements in soleus and EDL muscle-specific force production, fatigue resistance and partial recovery of cage ambulation found in the OVX-Li mice with low-dose Li supplementation could be clinically relevant for postmenopausal females, given that at least two-thirds of biological females undergoing menopause will experience fatigue (Boneva et al., 2015; Lee & Lee, 2020; O'Bryant et al., 2003; Taylor-Swanson

et al., 2018). In a cross-sectional study of 300 females, 85% of postmenopausal females and 46.5% of perimenopausal females reported symptoms of physical and mental exhaustion compared to just 19.7% of premenopausal females (Chedraui et al., 2007).

Nonetheless, we acknowledge several limitations of our study. First, although the OVX rodent model is a well-established and convenient preclinical tool for studying oestrogen deficiency, it does not entirely mimic the gradual hormonal changes seen in biological females during menopause. In this model, the ovaries are surgically removed, resulting in a sudden and substantial decrease in circulating oestrogen and ovarian hormones (Rodríguez-Landa, 2022; Souza et al., 2019). This approach enables precise control of experimental timing and minimizes confounding factors such as age, as OVX surgery can be performed in young adult animals as early as 12 weeks of age (Koebele & Bimonte-Nelson, 2016). However, it does not reflect the progressive decline in ovarian function characteristic of human perimenopause.

Alternative models of ovarian failure, such as the accelerated ovarian failure model induced by 4-vinylcyclohexene diepoxide (VCD), more closely approximate the gradual reduction in ovarian hormone production observed during natural menopausal transition (van Kempen et al., 2011). The VCD model selectively targets small primordial and primary follicles, resulting in a slower depletion of the ovarian reserve and a progressive decline in circulating oestrogen levels (Brooks et al., 2016; Koebele & Bimonte-Nelson, 2016; van Kempen et al., 2011). Interestingly, although VCD-induced gradual ovarian failure was shown to have minimal detrimental effects on *in vivo* muscle contractility and fatigue (Hinks et al., 2024), single fibres obtained from both the soleus and EDL muscles exhibit significant reductions in peak force (Hubbard et al., 2023). Future studies should therefore aim to replicate and expand on our findings of improved soleus and EDL contractility with Li treatment in the VCD model that more accurately reflect the gradual hormonal and physiological changes during the menopausal transition. These types of studies should also consider age. The mice used in this study were 26 weeks old at the time of ovariectomy and 28 weeks old at the time of Li intervention, corresponding to the stage of young adulthood. This is substantially younger than the age at which biological females typically experience menopause, and differences in baseline muscle mass, regenerative capacity and hormonal responsiveness between young adult and middle-aged animals may influence the observed outcomes.

It would also be advantageous to investigate various Li doses. The dose used in our present study has been shown to equate to a serum Li concentration of 0.15 mM (Marcella et al., 2024), which is below the therapeutic window typically prescribed to patients living with bipolar disorder (0.5–1.0 mM) (Hamstra et al., 2023). To enhance clinical translation, future work could assess the effects of Li supplementation within this therapeutic window in both OVX and VCD mouse models while also carefully monitoring murine health for signs of nephrotoxicity. Such studies could also explore a possible synergistic effect of Li supplementation combined with hormone replacement therapy (i.e. oestradiol treatment), particularly whether Li may lower the oestradiol dose needed to gain benefits for musculoskeletal and metabolic health. However, an alternative perspective is that our chosen dose for this study indicates that a subtherapeutic dose of Li, which likely offers a better safety margin, could still be effective for postmenopausal women. Therefore, based on our present findings, future clinical research at subtherapeutic doses should also be tested.

In conclusion, this study demonstrates that low-dose Li supplementation fully or partly restored skeletal muscle strength, endurance, cage activity, BMD and insulin tolerance in female OVX mice. Although the extent of these effects varied, given that lithium is already FDA-approved, our findings may reveal a viable therapeutic option for postmenopausal females. Future studies should investigate whether the therapeutic benefits of low-dose Li supplementation in OVX female mice can be applied to other preclinical menopause models, aiming to pave the path towards the clinical setting.

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

## Additional information

### Data availability statement

Data supporting the findings of this work can be made available upon reasonable request to the corresponding author.

### Competing interests

The authors declare that they have no competing interests.

### Author contributions

Conceptualization: B.M.M. and V.A.F. Methodology: B.M.M., J.L.B., B.L.H., M.S.G., A.A.M.T. and R.W.B. Investigation: B.M.M., J.L.B., B.L.H., M.S.G., A.A.M.T., R.W.B., R.V., R.E.K.M. and V.A.F. Visualization: B.M.M. and V.A.F. Supervision: R.V., R.E.K.M. and V.A.F. Writing (original draft): B.M.M. and V.A.F. Writing (review and editing): B.M.M., J.L.B., B.L.H., M.S.G., A.A.M.T., R.W.B., R.V., R.E.K.M. and V.A.F.

## Funding

This work was supported by a CIHR CRC Tier II Award to V.A.F. (CRC-2019-00412). B.M.M. was supported by a CIHR CGS-M award. J.L.B. was supported by a CIHR CGS-D award. B.L.H. was supported by a NSERC CGS-D award. A.A.M.T. was supported by a CIHR CGS-M award.

## Acknowledgements

The authors thank the Brock University animal care staff for their hard work and support, allowing for the completion of these projects.

## Keywords

fatigue, force production, GSK3, insulin resistance, oestrogen, osteoporosis, SERCA, skeletal muscle

## Supporting information

Additional supporting information can be found online in the Supporting Information section at the end of the HTML view of the article. Supporting information files available:

**Peer Review History**
**Supplementary Dataset 1**
**Supplementary Dataset 2**
**Supplementary Dataset 3**
**Supplementary Dataset 4**
**Supplementary Dataset 5**

