## [Peer Review History · The Journal of Physiology]

Low-dose lithium supplementation promotes musculoskeletal and metabolic health in ovariectomized female mice

Bianca M Marcella, Jessica L Braun, Amélie AT Marais, Briana Hockey, Mia S Geromella, Ryan W Baranowski, Rene Vandenboom, Rebecca EK MacPherson, and Val Andrew Fajardo
DOI: 10.1113/JP289990

Corresponding author(s): Val Andrew Fajardo (vfajardo@brocku.ca)

The following individual(s) involved in review of this submission have agreed to reveal their identity: Joachim Nielsen (Referee #2); Ken D O'Halloran (Referee #3)

Review Timeline:

Submission Date:	22-Aug-2025
Editorial Decision:	06-Oct-2025
Revision Received:	05-Dec-2025
Editorial Decision:	11-Dec-2025
Revision Received:	12-Dec-2025
Accepted:	16-Dec-2025

Senior Editor: Paul Greenhaff

Reviewing Editor: Robert Musci

Transaction Report:

Dear Dr Fajardo,

Re: JP-RP-2025-289990 "Low-dose lithium supplementation promotes musculoskeletal and metabolic health in ovariectomized female mice" by Bianca M Marcella, Jessica L Braun, Briana Hockey, Mia S Geromella, Amélie AT Marais, Ryan W Baranowski, Rene Vandenoorn, Rebecca EK MacPherson, and Val Andrew Fajardo

Thank you for submitting your manuscript to The Journal of Physiology. It has been assessed by a Reviewing Editor and by 3 expert referees and we are pleased to tell you that it is potentially acceptable for publication following satisfactory major revision.

REVISION CHECKLIST:

We look forward to receiving your revised submission.

Yours sincerely,

Paul Greenhaff
Senior Editor
The Journal of Physiology

REQUIRED ITEMS

- Author photo and profile. First or joint first authors are asked to provide a short biography (no more than 100 words for one author or 150 words in total for joint first authors) and a portrait photograph. These should be uploaded and clearly labelled together in a Word document with the revised version of the manuscript. See Information for Authors for further details.

- You must start the Methods section with a paragraph headed Ethical approval (https://jp.msubmit.net/cgi-bin/main.plex?form_type=display_requirements#methods).

Research must comply with The Journal's policies regarding animal experiments (<https://physoc.onlinelibrary.wiley.com/hub/animal-experiments>) and adherence to these policies must be stated in the manuscript.

Authors should confirm in their Methods section that their experiments were carried out according to the guidelines laid down by their institution's animal welfare committee, including an ethics approval reference number. The Methods section must contain a statement about access to food, water and housing, details of the anaesthetic regime: anaesthetic used, dose and route of administration, and method of killing the experimental animals.

- You must upload original, uncropped western blot/gel images (including controls) if they are not included in the manuscript. This is to confirm that no inappropriate, unethical or misleading image manipulation has occurred. These should be uploaded as 'Supporting information for review process only'. Please label/highlight the original gels so that we can clearly see which sections/lanes have been used in the manuscript figures. For more information, see: <https://physoc.onlinelibrary.wiley.com/hub/journal-policies#imagmanip>.

- Please ensure that any tables are editable and in Word format, and wherever possible, embedded in the article file itself.

- Papers must comply with the Statistics Policy: https://jp.msubmit.net/cgi-bin/main.plex?form_type=display_requirements#statistics.

In summary:

- If $n \leq 30$, all data points must be plotted in the figure in a way that reveals their range and distribution. A bar graph with data points overlaid, a box and whisker plot or a violin plot (preferably with data points included) are acceptable formats.

- If $n > 30$, then the entire raw dataset must be made available either as supporting information, or hosted on a not-for-profit repository, e.g. FigShare, with access details provided in the manuscript.

- 'n' clearly defined (e.g. x cells from y slices in z animals) in the Methods. Authors should be mindful of pseudoreplication.
- All relevant 'n' values must be clearly stated in the main text, figures and tables.
- The most appropriate summary statistic (e.g. mean or median and standard deviation) must be used. Standard Error of the Mean (SEM) alone is not permitted.
- Exact p values must be stated. Authors must not use 'greater than' or 'less than'. Exact p values must be stated to three significant figures even when 'no statistical significance' is claimed.

- Please include an Abstract Figure file, as well as the Figure Legend text within the main article file. The Abstract Figure is a piece of artwork designed to give readers an immediate understanding of the research and should summarise the main conclusions. If possible, the image should be easily 'readable' from left to right or top to bottom. It should show the physiological relevance of the manuscript so readers can assess the importance and content of its findings. Abstract Figures should not merely recapitulate other figures in the manuscript. Please try to keep the diagram as simple as possible and without superfluous information that may distract from the main conclusion(s). Abstract Figures must be provided by authors no later than the revised manuscript stage and should be uploaded as a separate file during online submission labelled as File Type 'Abstract Figure'. Please also ensure that you include the figure legend in the main article file. All Abstract Figures should be created using BioRender. Authors should use The Journal's premium BioRender account to export high-resolution images. Details on how to use and access the premium account are included as part of this email.

EDITOR COMMENTS

Reviewing Editor:

Ethics Concerns:

The surgery performed for OVX/OVX-sham was not described. JAX provides this as a service and they acquired the mice already ovariectomized. Detail on the surgical procedures may need to be provided per Journal standards.

Comments for Authors to ensure the paper complies with the Statistics Policy (Required):
Per Journal policy, In the figures, please provide specific p values. (For example in Figure 1)

Comments to the Author (Required):

Thank you for submitting your manuscript for consideration. Both reviewers commented positively on your manuscript. They also provided good recommendations to improve and enhance the clarity of your manuscript. Please address all comments.

The main concerns expressed regard methodology (ITT/GTT, contractility measures, and calcium uptake). There were also some concerns regarding statistics and inclusion of all mice for measures. Please be certain to address these concerns and clarify the reviewers' questions.

Finally, please refer to the Journal's statistical policy regarding the use of asterisks in figures.

Please also see 'Required Items' above.

Senior Editor:

Comments for Authors to ensure the paper complies with the Statistics Policy (Required):
Exact P values should be shown in figures.

Comments to the Author:

Thank you for the manuscript submission to The Journal of Physiology which has been considered by a reviewing editor and two expert reviewers. Both reviewers were relatively positive about the study but have raised a number of significant points that the authors are required address. Additionally, the Reviewing Editor and Reviewer 3 have raised important points regarding the need for a description of the surgical procedures to be included in the manuscript (as per Journal policy) and representation of statistical differences in the figures. We look forward to receiving the revised version of the manuscript

along with the authors' rebuttal.

REFeree COMMENTS

Referee #1:

The authors investigate whether low-dose lithium supplementation could mitigate musculoskeletal and metabolic impairments induced by OVX in female mice, a widely used model of "postmenopausal estrogen deficiency" (see general comment). They report that lithium treatment improved muscle contractile function (increased specific force in both soleus and EDL, improved fatigue resistance in the soleus), enhanced SERCA-mediated calcium uptake, reduced oxidative stress markers, and preserved bone mineral density. Additionally, lithium modestly improved insulin sensitivity, though it did not restore glucose tolerance, energy expenditure, or body composition. Their results indicate that lithium may confer partial protection against OVX-induced declines in muscle and bone health. Overall, the present study addresses an interesting and important question. The paper reads very clearly, the novelty is upfront, and the conclusions are well-supported by the data - it was a pleasure to read and provide feedback on. This would be a nice fit for the Journal.

General Point: OVX provides a convenient model of low levels of estrogen to study various physiological systems. An issue which often arises when using this model is attempting to draw parallels between OVX and the human condition of menopause. The authors don't over state this in the paper, but it may not be clear to the reader that other rodent models of estrogen deficiency exist which better approximate the natural change in hormones associated with ovarian failure. The authors may be familiar with the VCD model, if not, this is a model of gradual ovarian failure (DOI: 10.1016/j.brainres.2010.12.064). Additionally, some groups have reported minimal to no effects of ovarian failure on muscle contractility and fatigue in vivo (DOI: 10.1113/EP091735), and conflicting results at the cellular level (DOI: 10.1152/ajpccell.00318.2023). While I have no problems with the data reported in the paper under review, the authors could do a better job of positioning their findings in the context of studies using the OVX and VCD models.

Line 73-75: nice! Important to note.

Line 95-97: Dosing choice is unclear. The authors state that in order to avoid adverse effects in other tissues (please specify which), a therapeutic range of 0.5-1.0mM serum Li concentration is used. But the next few lines discuss a 0.02mM dose. It's unclear why those dosages were used if 0.5-1.0mM is still considered safe.

Line 153: Please specify if mice were fasted before ITT. Usually they are, but the following line says they were allowed access to food, which may impact results and interpretation.

Line 256: n=3 is a low sample size. What stats were used for this analysis?

Line 268: "Most statistical comparisons..." What do the authors mean by "most"? Were the data checked for normality before ANOVAs were run? Was the analysis powered?

Line 268: What were the factors in the 1way and 2way anovas?

In general for the results, reporting %differences would be a nice addition.

Fig1 caption: Bold "I"

Fig5 caption: Incorrect caption. Panel A-E show MHC fibre type distributions, but no mention in caption. MHC rep image of

SOL showed, not the EDL. Please specify why.

Fig 7 caption: Panel "B" not discussed

Line 315-317: The EDL results with respect to calcium uptake (Li group having higher calcium uptake than even sham) should be discussed in more detail, especially since Li group did not recover fatigue in the EDL.

The study says low dose LiCl,... is this the dose that would be prescribed to women? Is this a mode of treatment that would be prescribed instead of estrogen replacement?

Line 373: authors say mice were 28wks old when treatment of LiCl was started, but on line 126 (methods) they say mice were 26wks. Please correct and be consistent throughout.

Line 374-376: Unclear why the authors chose this "low" dosing, since they report that their dose is well below the therapeutic efficacy range of Li of 0.5-1.2 mM.

Line 427-429: May be true, but a big difference between the OVX mice used in this study and women who undergo menopause is age. A short discussion of how this may affect results may be helpful for the readership.

Line 437: Effects of the OVX surgery or estrogen deficiency?

Line 453: nice!

Line 478: You previously mentioned this was a very low dose compared to the therapeutic efficacy range. How would you confidently say the effects you saw with 10mg/kg/day is a result of the treatment? The dosage choice is unclear.

In general, how do you think the combination of E2 and LiCl would benefit post-menopausal women? A short discussion on this might help or even mentioned as a future direction (if not already discussed in the literature).

What are the effects of other doses of LiCl?

[END]

Referee #2:

The authors investigated the effects of low-dose lithium (Li) supplementation on muscle contractility, endurance, SR calcium handling, bone mineral density (BMD), and insulin sensitivity. Given the known metabolic effects of low-dose Li in males and its promising impact on osteogenesis, this study on female OVX mice has the potential for high impact in the field. The study is original and well-conducted. However, the conclusions may slightly overstate the findings, as not all group comparisons

show significant differences between the OVX-Li and OVX groups (see point 7 below). Therefore, the discussion and conclusion sections could be more balanced by acknowledging the uncertainty and limitations of the results. Impact on the area of research

1) What time of day were the GTT, ITT, and tissue collection performed? Given that several metabolic processes exhibit circadian variation (e.g., [10.1126/sciadv.abi9654]), this information is important for contextualizing the findings in relation to existing literature.

2) These points concern the contractility measures.

2A). Was the muscle length adjusted to an optimal length prior to testing? This is often defined as the resting length (where passive force begins) plus 10%, or the length at which maximal twitch force is observed.

2B). The interpretation of the EDL muscle data appears more complex than reported. The manuscript states, "No difference between groups in EDL fatigue resistance capacity (Fig. 2F)" (line 300), yet Figure 2E indicates a group effect. Could the authors clarify this? The time-course graphs suggest that OVX-Li retains force initially but then shows a more pronounced decline later. This pattern is not captured by the time to 50% force reduction metric. Why was 50% chosen, and not 30% or 70%? Could this reflect different fatigue mechanisms? For example, could the initial resistance be due to higher CrP levels in OVX-Li, followed by accelerated fatigue due to Pi accumulation or SR Ca-Pi precipitation? It is acceptable if the authors choose not to speculate, but clarification would be helpful.

2C). The force-frequency curves are interpreted as showing reduced force in OVX mice (Fig. 2A and 2D), but the significant frequency \times group interaction suggests that force production is also frequency-dependent. Would it be relevant to calculate the frequency required to reach 50% maximal force (F50%)? A group difference in F50% could indicate altered myofibrillar calcium sensitivity. Additionally, comparing twitch to tetanic force might provide insight into SR Ca²⁺ release. If twitch force is more affected than tetanic force, this could suggest impaired SR Ca²⁺ release.

3) Was the dose-response of the western blots adjusted to account for potential non-linear relationships between signal intensity and protein quantity?

4) These points concern the SR Ca²⁺ uptake measures

4A) Please clarify what "LoggerPro" refers to (line 219).

4B) The text states that uptake rates are expressed as $\mu\text{mol Ca}^{2+}/\text{g protein}/\text{min}$, but Figure 4 uses $\text{nmol}/\text{g}/\text{min}$. Which is correct?

4C) The manuscript mentions measurements at 500 and 1000 nM Ca²⁺, but Figure 4 shows 500 and 250 nM. Please clarify.

4D) Since calcium uptake rate depends on free Ca²⁺ concentration, would it be informative to also report the time required to take up 50% of the calcium? This could provide complementary information to the instantaneous uptake rate.

5) The statistical section (line 269) mentions outlier removal. While the cohort size is reported as 9-10 mice, many results are based on 3-7 observations. Is the reduced sample size due to outlier exclusion or were only subsets of animals analysed?

6) In Figure 7A, it is indicated that the sham group differs from OVX-Li. However, this is not visually apparent. Could the authors clarify?

7) Some findings are not clearly distinct and this should be acknowledged in the discussion. For example, soleus endurance is not significantly different between OVX and OVX-Li (only compared to sham); cage ambulation, 4-HNE content, and ITT results for OVX-Li are not significantly different from either sham or OVX. The discussion should reflect this uncertainty. One suggestion is to revise the summary paragraph (lines 363-373) to better represent the overall phenotype, highlighting the more robust effects of Li while acknowledging the variability and limitations in the data.

Referee #3 (ethics review):

Thank you for submitting your manuscript to The Journal of Physiology. Additional details pertaining to animal ethics and welfare are required.

1. Please confirm that your study adheres to the ethics principles and policies of The Journal and ensure that you are compliant with reporting requirements detailed in our recent Editorial; see <https://physoc.onlinelibrary.wiley.com/doi/10.1113/JP286666>

2. Line 122. Although OVX mice were provided by a registered commercial supplier, we require details on surgical procedures performed to satisfy our requirement to ensure that the highest standards of care were adhered to. Please provide details on anaesthetic agents used, including dose and route; how an adequate depth of anaesthesia was confirmed during procedures; and, details on post-procedural care.

3. You must start the Methods section with the subheading "Ethical approval".

4. Line 160. Give details of the anaesthetic agent(s) used, the dose and route of administration, and how an adequate depth of anaesthesia was confirmed before proceeding to the method of killing.

5. Line 138. I suggest moving this section to later in the methods narrative as you indicate it was performed at study endpoint. Please elaborate on this procedure if mice were alive (eg anaesthesia details), or confirm that mice were killed (restating the method) before measurements were taken.

END OF COMMENTS

Response to Reviewers – JP-RP-2025-289990

General Response to Reviewers:

We'd like to thank the reviewers for their insightful comments. We have carefully addressed each of the points. For clarity, the reviewers' comments have been *italicized*, and our responses are in **bold** font. With the following responses to the reviewers' concerns and corresponding changes to our manuscript, we are hopeful that our review paper will be suitable for publication.

EDITOR COMMENTS

Ethics Concerns:

The surgery performed for OVX/OVX-sham was not described. JAX provides this as a service and they acquired the mice already ovariectomized. Detail on the surgical procedures may need to be provided per Journal standards.

*Comments for Authors to ensure the paper complies with the Statistics Policy (Required):
Per Journal policy, In the figures, please provide specific p values. (For example in Figure 1)*

Comments to the Author (Required):

Thank you for submitting your manuscript for consideration. Both reviewers commented positively on your manuscript. They also provided good recommendations to improve and enhance the clarity of your manuscript. Please address all comments. The main concerns expressed regard methodology (ITT/GTT, contractility measures, and calcium uptake). There were also some concerns regarding statistics and inclusion of all mice for measures. Please be certain to address these concerns and clarify the reviewers' questions.

Finally, please refer to the Journal's statistical policy regarding the use of asterisks in figures.

Please also see 'Required Items' above.

Senior Editor:

*Comments for Authors to ensure the paper complies with the Statistics Policy (Required):
Exact P values should be shown in figures.*

Comments to the Author:

Thank you for the manuscript submission to The Journal of Physiology which has been considered by a reviewing editor and two expert reviewers. Both reviewers were relatively positive about the study but have raised a number of significant points that the authors are required address. Additionally, the Reviewing Editor and Reviewer 3 have raised important points regarding the need for a description of the surgical procedures to be included in the manuscript (as per Journal policy) and representation of statistical differences in the figures. We look forward to receiving the revised version of the manuscript along with the authors' rebuttal.

Thank you for your positive feedback. We have thoroughly addressed all reviewers' concerns. Additionally, we contacted JAX to obtain detailed information on the anesthetic agents used, including dose and route of administration, confirmation of adequate anesthetic depth during procedures, and post-procedural care. All figures now include exact p-values, except for the RNA-seq figure (Figure 7), where displaying exact P-values

for the top 50 differentially expressed genes would make the heat map visually unclear.

REFEREE COMMENTS

Referee #1:

The authors investigate whether low-dose lithium supplementation could mitigate musculoskeletal and metabolic impairments induced by OVX in female mice, a widely used model of "postmenopausal estrogen deficiency" (see general comment). They report that lithium treatment improved muscle contractile function (increased specific force in both soleus and EDL, improved fatigue resistance in the soleus), enhanced SERCA-mediated calcium uptake, reduced oxidative stress markers, and preserved bone mineral density. Additionally, lithium modestly improved insulin sensitivity, though it did not restore glucose tolerance, energy expenditure, or body composition. Their results indicate that lithium may confer partial protection against OVX-induced declines in muscle and bone health. Overall, the present study addresses an interesting and important question. The paper reads very clearly, the novelty is upfront, and the conclusions are well-supported by the data - it was a pleasure to read and provide feedback on. This would be a nice fit for the Journal.

General Point: OVX provides a convenient model of low levels of estrogen to study various physiological systems. An issue which often arises when using this model is attempting to draw parallels between OVX and the human condition of menopause. The authors don't over state this in the paper, but it may not be clear to the reader that other rodent models of estrogen deficiency exist which better approximate the natural change in hormones associated with ovarian failure. The authors may be familiar with the VCD model, if not, this is a model of gradual ovarian failure (DOI: 10.1016/j.brainres.2010.12.064). Additionally, some groups have reported minimal to no effects of ovarian failure on muscle contractility and fatigue in vivo (DOI: 10.1113/EP091735), and conflicting results at the cellular level (DOI: 10.1152/ajpcell.00318.2023). While I have no problems with the data reported in the paper under review, the authors could do a better job of positioning their findings in the context of studies using the OVX and VCD models.

We fully agree that the OVX model, although widely used and convenient experimentally, does not represent the gradual hormonal changes that occur during the perimenopausal transition in women. We have now added a Limitations section to better contextualize our findings within the broader spectrum of estrogen-deficiency models, including discussion of the VCD model, as well as limitations regarding the age of the mice used in this study, the Li dose tested, and future directions (L725-764).

Line 73-75: nice! Important to note.

Agreed!

Line 95-97: Dosing choice is unclear. The authors state that in order to avoid adverse effects in other tissues (please specify which), a therapeutic range of 0.5-1.0mM serum Li concentration is used. But the next few lines discuss a 0.02mM dose. It's unclear why those dosages were used if 0.5-1.0mM is still considered safe.

This reviewer raises a very important point, one also acknowledged by reviewer 2. We have now added information justifying our dose choice in both the introduction (L95-117) and discussion (L480-486). Essentially, in mice, Li treatment within the range of 0.4-0.8 mM has been shown to cause adverse effects, particularly on the kidneys, making the use of lower doses essential (Nespital et al., PMID: 34532960).

Line 153: Please specify if mice were fasted before ITT. Usually they are, but the following line says they were allowed access to food, which may impact results and interpretation.

We apologize for the confusion. The mice were not fasted before ITT. In our lab, mice are fasted before a glucose tolerance test (GTT) to measure the response to a defined glucose bolus from a stable, comparable baseline. However, we view that fasting prior to ITT is not required and actually increases the risk of hypoglycemia. Moreover, a recent study found that insulin's ability to lower blood glucose (i.e., insulin responsiveness) during an ITT was similar across 0-, 2-, 4-, and 6-h fasts in mice (PMID: 32739449).

Line 256: n=3 is a low sample size. What stats were used for this analysis?

We acknowledge that an n = 3 per group represents a relatively small sample size. However, this is common practice in RNA-seq studies due to the high cost of sequencing and bioinformatic analysis. Differential expression analysis was performed by Novogene using the DESeq2 package, which models normalized count data with a Negative Binomial Distribution and applies the Benjamini–Hochberg (BH) method to control for multiple comparisons. DESeq2 specifically accounts for small sample sizes by borrowing information across all genes to estimate dispersion parameters, thereby improving variance estimation and statistical power (Love et al., 2014; PMID: 25516281). These details are now found in the RNA-Seq section (L293-303).

Line 268: "Most statistical comparisons..." What do the authors mean by "most"? Were the data checked for normality before ANOVAs were run? Was the analysis powered? Line 268: What were the factors in the 1way and 2way anovas?

We thank the reviewer for this comment. We have revised the Methods section to describe the factors included in each ANOVA test explicitly and to clarify how normality was assessed (L306-317).

Normality of all datasets was verified using the Shapiro–Wilk test prior to running parametric analyses. When data were not normally distributed, the non-parametric Kruskal–Wallis test was used instead of a one-way ANOVA (for EDL: GSK3, pGSK3:GSK3 ratio, time to 90% initial force and CS content).

Power calculations were based on contractility data from a previous study in male mice treated with lithium (PMID: 32729236), which indicated that $n = 10$ per group would provide sufficient power ($\beta = 0.8$, $\alpha = 0.05$) to detect physiologically relevant differences in force production. This served as an estimate, and in some instances, significant effects were detected with fewer than 10 samples per group.

In general, for the results, reporting %differences would be a nice addition.

Throughout the results section, we now report % differences and fold-changes when describing our findings.

Fig1 caption: Bold "I"

This has been done.

Fig5 caption: Incorrect caption. Panel A-E show MHC fibre type distributions, but no mention in caption. MHC rep image of SOL showed, not the EDL. Please specify why.

We have now updated the caption for Figure 5 to include the description of panels A-E. Additionally, we have added Figure 6, which presents similar fibre type and western blot analysis in the EDL.

Fig 7 caption: Panel "B" not discussed

We appreciate the reviewer for bringing this to our attention. It has been corrected. Please note that this figure is now labeled as Figure 8 in our revised submission.

Line 315-317: The EDL results with respect to calcium uptake (Li group having higher calcium uptake than even sham) should be discussed in more detail, especially since Li group did not recover fatigue in the EDL.

Results from our EDL calcium uptake experiments are now discussed in relation to our EDL contractile experiments, highlighting that the OVX-Li group had higher calcium uptake than even sham (L563-568). We also highlight potential mechanisms using findings derived from RNA-Seq, specifically how increased expression of *Atp2a2*, *Casq1* and *Casq2* with Li supplementation might contribute to the enhanced calcium uptake found in the OVX-Li muscle (L621-623).

Please note, that in our original submission, we made an error in our figure. The reported “group effect” in the EDL fatigue curve was actually a significant group \times time interaction. As predicted by reviewer 2, this interaction indicates a transient difference in EDL muscle fatigue between OVX-Li and OVX groups within the first 16 s of the protocol—approximately when force declines from 100% to 90% of the initial value. Further analysis shows that OVX-Li EDL muscles, on average, take longer to reach 90% of initial force compared with OVX muscles. We discuss this finding in relation to calcium uptake and other RNA-Seq findings in our discussion.

The study says low-dose LiCl,... is this the dose that would be prescribed to women? Is this a mode of treatment that would be prescribed instead of estrogen replacement?

These are good questions. In our discussion, we mention that doses below and within the therapeutic range typically used in the clinical setting for bipolar disorder should be tested. We also highlight the importance of future studies that determine whether Li can be used in place of or perhaps alongside HRT (L752-764).

Line 373: authors say mice were 28wks old when treatment of LiCl was started, but on line 126 (methods) they say mice were 26wks. Please correct and be consistent throughout.

The mice were 28 weeks old when Li treatment started. This has been corrected on L480. We have also added more details to the methods section (L136-150).

Line 374-376: Unclear why the authors chose this "low" dosing, since they report that their dose is well below the therapeutic efficacy range of Li of 0.5-1.2 mM.

We apologize for the confusion. The term ‘efficacy’ should not have been included when mentioning the therapeutic range typically used for bipolar disorder as it makes it seem as though treatment below these levels would not be effective. As outlined in our introduction, treatment at ‘subtherapeutic Li’ levels (based on those typically used for bipolar) has still been shown to be effective in rodent models (L99-117). Furthermore, since it has been previously demonstrated that mice subject to the therapeutic range prescribed to humans still exhibit signs of kidney toxicity (Nespital et al., PMID: 34532960), we opted for a Li dose that would be below this range of 0.5-1.2 mM. These changes are also reflected in L480-486.

Line 427-429: May be true, but a big difference between the OVX mice used in this study and women who undergo menopause is age. A short discussion of how this may affect results may be helpful for the readership.

Agreed. We have addressed this comment in the new limitations section at the end of our discussion (L746-751).

Line 437: Effects of the OVX surgery or estrogen deficiency?

This is a good question. Since OVX surgery also causes declines in other hormones like progesterone, we have referred to it as OVX surgery rather than just estrogen deficiency.

Line 453: nice!

Thank you!

Line 478: You previously mentioned this was a very low dose compared to the therapeutic efficacy range. How would you confidently say the effects you saw with 10mg/kg/day is a result of the treatment? The dosage choice is unclear.

We again apologize for the confusion, as the term ‘efficacy’ should not have been included when describing the therapeutic range. Instead, when discussing the therapeutic range, we meant to describe the dose typically used for bipolar therapy—which is the most common use of lithium—being maintained within a narrow range (serum Li 0.5-1.2 mM) to minimize adverse side effects. Throughout the literature, there are several studies in rodents still showing positive benefits of Li at doses below this range of concentrations.

In general, how do you think the combination of E2 and LiCl would benefit post-menopausal women? A short discussion on this might help or even mentioned as a future direction (if not already discussed in the literature). What are the effects of other doses of LiCl?

These are good questions, and we’ve included a discussion in our limitations and future directions section (L752-760).

Referee #2:

The authors investigated the effects of low-dose lithium (Li) supplementation on muscle contractility, endurance, SR calcium handling, bone mineral density (BMD), and insulin sensitivity. Given the known metabolic effects of low-dose Li in males and its promising impact on osteogenesis, this study on female OVX mice has the potential for high impact in the field. The study is original and well-conducted. However, the conclusions may slightly overstate the findings, as not all group comparisons show significant differences between the OVX-Li and OVX groups (see point 7 below). Therefore, the discussion and conclusion sections could be more balanced by acknowledging the uncertainty and limitations of the results. Impact on the area of research.

We thank the reviewer for their positive and constructive feedback. We have carefully revised our manuscript to better describe our results and to present a more balanced discussion.

1) What time of day were the GTT, ITT, and tissue collection performed? Given that several metabolic processes exhibit circadian variation (e.g., [10.1126/sciadv.abi9654]), this information is important for contextualizing the findings in relation to existing literature.

We agree with the reviewer and have included the following details in our methods. Both GTT and ITT were carried out starting at 10:00h (L183-184). All tissue collection occurred between 10:00 h and 14:00 h, with subjects sampled one at a time in alternating order (sham, OVX, OVX-Li) (L193-195).

2) These points concern the contractility measures.

2A). Was the muscle length adjusted to an optimal length prior to testing? This is often defined as the resting length (where passive force begins) plus 10%, or the length at which maximal twitch force is observed.

Yes, “Muscle length was first set to the optimal length, determined as the length at which the maximal isometric twitch force was observed.” L200-201

2B). The interpretation of the EDL muscle data appears more complex than reported. The manuscript states, "No difference between groups in EDL fatigue resistance capacity (Fig. 2F)" (line 300), yet Figure 2E indicates a group effect. Could the authors clarify this? The time-course graphs suggest that OVX-Li retains force initially but then shows a more pronounced decline later. This pattern is not captured by the time to 50% force reduction metric. Why was 50% chosen, and not 30% or 70%? Could this reflect different fatigue mechanisms? For example, could the initial resistance be due to higher CrP levels in OVX-Li, followed by accelerated fatigue due to Pi accumulation or SR Ca-Pi precipitation? It is acceptable if the authors choose not to speculate, but clarification would be helpful.

We thank the reviewer for this insightful comment. First, there was an error in the figure: the reported “group effect” was actually a significant $group \times time$ interaction. This indicates a transient difference in EDL muscle fatigue between OVX-Li and OVX groups

within the first 20 s of the protocol—approximately when force declines from 100% to 90% of the initial value. Further analysis shows that OVX-Li EDL muscles, on average, take longer to reach 90% of initial force compared with OVX muscles. However, as the reviewer correctly noted, this difference disappears by the time muscles reach 50% of their initial force. These corrections and clarifications are now included in the revised Results section (L358-364) and in Figure 2, indicating that Li did indeed have an acute and positive effect on EDL fatigue resistance.

Similarly, in the soleus muscle, the observed difference at the time to 70% of initial force was no longer apparent at 50%. In the revised Results, we now provide a clearer rationale for using 70% threshold for the soleus (L348-354). Essentially, we state that these points of comparison were chosen based on the significant *group* × *time* interactions, which indicated transient differences early in the fatigue protocol.

Finally, regarding the potential mechanisms of fatigue, we appreciate the reviewer's insightful suggestions and have included additional discussion to address the possibility that initial resistance in OVX-Li muscles may relate to altered CrP dynamics (L588-599).

2C). The force-frequency curves are interpreted as showing reduced force in OVX mice (Fig. 2A and 2D), but the significant frequency × group interaction suggests that force production is also frequency-dependent. Would it be relevant to calculate the frequency required to reach 50% maximal force (F50%)? A group difference in F50% could indicate altered myofibrillar calcium sensitivity. Additionally, comparing twitch to tetanic force might provide insight into SR Ca²⁺ release. If twitch force is more affected than tetanic force, this could suggest impaired SR Ca²⁺ release.

We thank the reviewer for this insightful suggestion and we have now analyzed and included the twitch:tetanus ratio ($P_t:P_o$), which provides complementary information about SR Ca²⁺ release and/or myofilament Ca²⁺ sensitivity. Our results show that the OVX soleus has a significantly lower $P_t:P_o$ compared with OVX-Li and sham soleus muscles; however, there were no differences in the EDL (see revised Figure 2).

As such, our results indicate that the force-enhancing effect in the soleus, but not the EDL, may be attributed to an enhancement in SR Ca²⁺ release and/or myofilament Ca²⁺ sensitivity; however, future studies using more dedicated experiments aimed at assessing myofilament Ca²⁺ sensitivity using permeabilized fibres are needed and could still potentially reveal an effect in both muscle types.

3) Was the dose-response of the western blots adjusted to account for potential non-linear relationships between signal intensity and protein quantity?

Protein loading amounts were optimized to prevent saturation of signal intensities.

4) *These points concern the SR Ca²⁺ uptake measures*
4A) *Please clarify what "LoggerPro" refers to (line 219).*

Apologies: LoggerPro is analytical software used for graphing and measuring instantaneous velocities (tangent analysis) in calcium uptake curves. We have corrected this in L264.

4B) *The text states that uptake rates are expressed as $\mu\text{mol Ca}^{2+}/\text{g protein}/\text{min}$, but Figure 4 uses $\text{nmol}/\text{g}/\text{min}$. Which is correct?*

We thank the reviewer for catching this. The correct units are displayed in the Figure ($\text{nmol}/\text{g}/\text{min}$) and this has been corrected in the methods (L267).

4C) *The manuscript mentions measurements at 500 and 1000 nM Ca²⁺, but Figure 4 shows 500 and 250 nM. Please clarify.*

The Figure is accurate and we have fixed our methods to say that measurements were made at 250 and 500 nM Ca²⁺. L266

4D) *Since calcium uptake rate depends on free Ca²⁺ concentration, would it be informative to also report the time required to take up 50% of the calcium? This could provide complementary information to the instantaneous uptake rate.*

We thank the reviewer for this insightful suggestion. While the time to 50% calcium uptake can indeed provide complementary information, we chose not to include this because it is not normalized to total protein content and therefore does not allow for fair comparisons between samples. In contrast, the instantaneous calcium uptake rates, which account for the effect of free Ca²⁺ on calcium uptake, are normalized to total protein, providing a more reliable and quantitative measure of calcium handling capacity across experimental groups.

5) *The statistical section (line 269) mentions outlier removal. While the cohort size is reported as 9-10 mice, many results are based on 3-7 observations. Is the reduced sample size due to outlier exclusion or were only subsets of animals analysed?*

We appreciate the reviewer's comment and the opportunity to clarify this point. The reduced sample sizes in some analyses reflect a combination of both outlier exclusion and the use of subsets of animals for specific assays. The reported cohort size of 9–10 mice applies to our *in vivo* measurements (e.g., DEXA scanning and insulin tolerance tests). In our contractility experiments, a few data points were excluded due to technical issues—such as compromised muscle integrity during dissections—which resulted in non-functional samples in addition to finding statistical outliers. For our biochemical assays (including calcium uptake, western blotting, and fibre typing), we typically begin with smaller sample sizes before expanding to larger cohorts, if needed.

6) In Figure 7A, it is indicated that the sham group differs from OVX-Li. However, this is not visually apparent. Could the authors clarify?

We apologize for the confusion. The asterisks were intended to be placed between OVX and OVX-Li, not OVX-Li and sham. We have corrected this and also displayed the exact p-values on the revised Figure 8A.

7) Some findings are not clearly distinct and this should be acknowledged in the discussion. For example, soleus endurance is not significantly different between OVX and OVX-Li (only compared to sham); cage ambulation, 4-HNE content, and ITT results for OVX-Li are not significantly different from either sham or OVX. The discussion should reflect this uncertainty. One suggestion is to revise the summary paragraph (lines 363-373) to better represent the overall phenotype, highlighting the more robust effects of Li while acknowledging the variability and limitations in the data.

We thank the reviewer for this helpful comment. We agree that some findings are less distinct and have revised the summary paragraph and discussion to reflect the varying degrees with which Li had an effect. Specifically, the revised text now highlights the more robust effects of Li supplementation that statistically differed between OVX-Li and OVX groups, while also acknowledging measures where OVX-Li mice were not statistically different from sham, representing moderate or partial improvements compared with OVX alone (L468-479).

We also apologize for the confusion in our earlier description—we now state that soleus and EDL muscle endurance was improved and was significantly different between OVX and OVX-Li mice.

Overall, we have carefully revised our results and discussion to reflect our findings more accurately.

Referee #3 (ethics review):

Thank you for submitting your manuscript to The Journal of Physiology. Additional details pertaining to animal ethics and welfare are required.

1. Please confirm that your study adheres to the ethics principles and policies of The Journal and ensure that you are compliant with reporting requirements detailed in our recent Editorial; see <https://physoc.onlinelibrary.wiley.com/doi/10.1113/JP286666>

We confirm that our study adheres to the ethics principles and policies of the Journal and that we are compliant with reporting requirements.

2. Line 122. Although OVX mice were provided by a registered commercial supplier, we require details on surgical procedures performed to satisfy our requirement to ensure that the highest standards of care were adhered to. Please provide details on anaesthetic agents used, including dose and route; how an adequate depth of anaesthesia was confirmed during procedures; and, details on post-procedural care.

We appreciate the reviewer's comment and have added further details in the Methods section (L136-148). The OVX mice used came from The Jackson Laboratory (JAX, Bar Harbor, ME, USA), an AAALAC-accredited facility. All surgical procedures performed at JAX are approved by their IACUC and follow aseptic, atraumatic techniques. According to JAX guidelines, mice were anesthetized with inhaled isoflurane, and carprofen was given for pain relief. Proper anesthesia depth was confirmed by the absence of pedal and palpebral reflexes before incision and maintained throughout the surgery. During anesthesia and recovery, supplemental heat was provided. After surgery, animals received subcutaneous sterile saline warmed and were monitored continuously until they could walk freely, then daily until their wounds healed completely (generally 7–10 days). Wound clips were removed prior to shipment.

3. You must start the Methods section with the subheading "Ethical approval".

This has been done.

4. Line 160. Give details of the anaesthetic agent(s) used, the dose and route of administration, and how an adequate depth of anaesthesia was confirmed before proceeding to the method of killing.

The following has now been included in our methods (L187-189).

Following the 8-week study, mice were euthanized by exsanguination under vaporized isoflurane anesthesia (5% in oxygen at a flow rate of 1–2 L/min). Adequate anesthetic depth was confirmed by the absence of a pedal withdrawal (toe pinch) reflex prior to proceeding.

5. Line 138. I suggest moving this section to later in the methods narrative as you indicate it was performed at study endpoint. Please elaborate on this procedure if mice were alive (eg anaesthesia details), or confirm that mice were killed (restating the method) before measurements were taken.

We apologize for the confusion. This was not the endpoint of the study and mice were not killed at this timepoint. We have provided additional clarification in our methods (L161-165).

“The iNSiGHT Small Animal dual-energy x-ray absorptiometry (DXA) Scanner (Scintica Instrumentation, ON, Canada) was used to measure total body mass, fat mass, lean mass, and BMD at the end of the 8-week treatment schedule. Mice were anesthetized under vaporized isoflurane (induction: 5% in oxygen at a flow rate of 1–2 L/min) and then positioned in the DXA scanner, which was fitted with tubing to deliver vaporized isoflurane at a maintenance concentration of 0.25% in oxygen (flow rate: 1–2 L/min). After the scan (~35 s), mice were allowed to recover fully before being returned to their home cages.”

Dear Dr Fajardo,

Re: JP-RP-2025-289990R1 "Low-dose lithium supplementation promotes musculoskeletal and metabolic health in ovariectomized female mice" by Bianca M Marcella, Jessica L Braun, Amélie AT Marais, Briana Hockey, Mia S Geromella, Ryan W Baranowski, Rene Vandenboom, Rebecca EK MacPherson, and Val Andrew Fajardo

Thank you for submitting your manuscript to The Journal of Physiology. It has been assessed by a Reviewing Editor and by 3 expert referees and we are pleased to tell you that it is acceptable for publication following satisfactory revision.

REVISION CHECKLIST:

Please upload two versions of your manuscript text: one with all relevant changes highlighted and one clean version with no changes tracked. The manuscript file should include all tables and figure legends, but each figure/graph should be uploaded as separate, high-resolution files. The journal is now integrated with Wiley's Image Checking service. For further details, see: <https://www.wiley.com/en-us/network/publishing/research-publishing/trending-stories/upholding-image-integrity-wileys->

image-screening-service

We look forward to receiving your revised submission.

Yours sincerely,

Paul Greenhaff
Senior Editor
The Journal of Physiology

REQUIRED ITEMS

- Papers must comply with the Statistics Policy: https://jp.msubmit.net/cgi-bin/main.plex?form_type=display_requirements#statistics.

In summary:

- If $n \leq 30$, all data points must be plotted in the figure in a way that reveals their range and distribution. A bar graph with data points overlaid, a box and whisker plot or a violin plot (preferably with data points included) are acceptable formats.
- If $n > 30$, then the entire raw dataset must be made available either as supporting information, or hosted on a not-for-profit repository, e.g. FigShare, with access details provided in the manuscript.
- 'n' clearly defined (e.g. x cells from y slices in z animals) in the Methods. Authors should be mindful of pseudoreplication.
- All relevant 'n' values must be clearly stated in the main text, figures and tables.
- The most appropriate summary statistic (e.g. mean or median and standard deviation) must be used. Standard Error of the Mean (SEM) alone is not permitted.
- Exact p values must be stated. Authors must not use 'greater than' or 'less than'. Exact p values must be stated to three significant figures even when 'no statistical significance' is claimed.

EDITOR COMMENTS

Reviewing Editor:

Thank you for taking the time to make revisions to your manuscript. As you will find, the reviewers agree that the revisions have sufficiently addressed their concerns.

However, please note Reviewer 2's comment that Figure 4 is missing half the data: "One note is that Figure 4 does not appear to be included in its full size: only the data from the soleus are shown, with the EDL data missing." Please ensure the proper figure/file is uploaded.

Please also see 'Required Items' above.

Senior Editor:

Comments for Authors to ensure the paper complies with the Statistics Policy (Required):

The Journal of Physiology requirement for authors to provide exact P values has not been met. The authors are using * $p < 0.05$, ** $p < 0.01$,

*** $p < 0.001$, and **** $p < 0.0001$ in the figures rather than exact P values. Please amend.

Comments to the Author:

Thank you for the revised manuscript. The Reviewing Editor and two expert reviewers are of the opinion that the manuscript has been improved and their comments have been addressed. There does appear to be an issue with the Figures (see the comment from the Reviewing Editor). Furthermore, The Journal of Physiology requirement for authors to provide exact P values in Results and Figures has not been met. The authors appear to be currently using * $p < 0.05$, ** $p < 0.01$,

*** $p < 0.001$, and **** $p < 0.0001$ throughout (including Figures), rather than exact P values. Please amend. We look forward to receiving the revised version of the manuscript. Thank you.

REFEREE COMMENTS

Referee #1:

All of my initial concerns/comments have been addressed.

Well done.

Referee #2:

The authors have adequately addressed my comments and questions. One note is that Figure 4 does not appear to be included in its full size: only the data from the soleus are shown, with the EDL data missing.

Referee #3:

Thank you for addressing the previous round of comments and for making revisions to the text. All points have been satisfactorily addressed. There are no further issues.

END OF COMMENTS

Response to Reviewers – JP-RP-2025-289990

General Response to Reviewers and Editor:

We'd like to thank the reviewers and the senior editor for the time they spent evaluating our paper; we believe their insightful comments have strengthened the manuscript. We have now included the correct Figure 4 and have removed the statements $*p < 0.05$, $**p < 0.01$, $***p < 0.001$, and $****p < 0.0001$ from the legends of Figures 1 and 4, as we now display exact p-values instead.

Dear Dr Fajardo,

Re: JP-RP-2025-289990R2 "Low-dose lithium supplementation promotes musculoskeletal and metabolic health in ovariectomized female mice" by Bianca M Marcella, Jessica L Braun, Amélie AT Marais, Briana Hockey, Mia S Geromella, Ryan W Baranowski, Rene Vandenboom, Rebecca EK MacPherson, and Val Andrew Fajardo

We are pleased to tell you that your paper has been accepted for publication in The Journal of Physiology.

Yours sincerely,

Paul Greenhaff
Senior Editor
The Journal of Physiology

IMPORTANT POINTS TO NOTE FOLLOWING ACCEPTANCE OF YOUR PAPER:

- **IMPORTANT NOTICE ABOUT OPEN ACCESS:** To assist authors whose funding agencies mandate immediate public access to published research findings, The Journal of Physiology allows authors to pay an Open Access (OA) fee to have their papers made freely available immediately on publication.

- You can help your research get the attention it deserves! Check out Wiley's free Promotion Guide for best-practice recommendations for promoting your work at: www.wileyauthors.com/eoo/guide. You can learn more about Wiley Editing Services which offers professional video, design, and writing services to create shareable video abstracts, infographics, conference posters, lay summaries, and research news stories for your research at: www.wileyauthors.com/eoo/promotion.

- If you would like to receive our 'Research Roundup', a monthly newsletter highlighting the cutting-edge research published in The Physiological Society's family of journals (The Journal of Physiology, Experimental Physiology, Physiological Reports, The Journal of Nutritional Physiology and The Journal of Precision Medicine: Health and Disease), please click this link, fill in your name and email address and select 'Research Roundup': <https://www.physoc.org/journals-and-media/membernews>

EDITOR COMMENTS

Reviewing Editor:

Thank you for making further revisions to the manuscript.

Senior Editor:

Thank you for making the further amendments to ensure the manuscript complies with Journal of Physiology statistical guidelines. The manuscript is now acceptable for publication. Congratulations.